# Collaborative Device-Cloud LLM Inference through Reinforcement Learning

## Abstract

Device-cloud collaboration has emerged as a promising paradigm for deploying large language models (LLMs), combining the efficiency of lightweight on-device inference with the superior performance of powerful cloud LLMs. An essential problem in this scenario lies in deciding whether a given query is best handled locally or delegated to the cloud. Existing approaches typically rely on external routers, implemented as binary classifiers, which often struggle to determine task difficulty from the prompt's surface pattern. To address these limitations, we propose a framework where the on-device LLM makes routing decisions at the end of its solving process, with this capability instilled through post-training. In particular, we formulate a reward maximization problem with carefully designed rewards that encourage effective problem solving and judicious offloading to the cloud. To solve this problem, we develop a group-adaptive policy gradient algorithm, featuring a group-level policy gradient, designed to yield an unbiased gradient estimator of the reward, and adaptive prompt filtering, developed to provide complementary learning signals. Extensive experiments across models and benchmarks show that the proposed methodology consistently outperforms existing baselines and significantly narrows the gap to full cloud LLM performance.

## 1 Introduction

Large language models (LLMs) have achieved remarkable success across a wide range of tasks, and LLM APIs are used in a variety of scenarios owing to their promising performance (Touvron et al., 2023; Achiam et al., 2023). Typically, these APIs operate under a cloud-based paradigm: user queries are transmitted to powerful LLMs hosted on cloud servers. While effective, this architecture places a heavy computational burden on the cloud, introduces non-negligible communication overhead for users, especially in bandwidth-constrained environments, and does not utilize the potential of local computation (Jin & Wu, 2024).

To alleviate these limitations, recent research has explored on-device LLMs, smaller models optimized for deployment on user devices (Liu et al., 2024; Xu et al., 2024; Fang et al., 2025). However, due to their limited parameter volume, the lightweight on-device models often lag behind cloud LLMs in terms of performance. This creates a critical trade-off between efficiency and accuracy.

To overcome this efficiency-accuracy trade-off, recent works have proposed collaborative frameworks that combine on-device and cloud LLMs (He et al., 2024; Li et al., 2025). Within these frameworks, a local router, typically implemented as a separate classifier, decides whether a request should be processed by the on-device LLM or offloaded to the cloud (Ong et al., 2025). This design leverages the efficiency of local inference (e.g., computation, communication) while retaining access to the superior performance of the cloud LLM. However, the router itself requires additional task-specific training on top of the on-device LLM's post-training (Ding et al., 2024; Yuan et al., 2025), which increases system complexity and restricts adaptability across tasks. Moreover, decoupling routing from the on-device LLM prevents the model from jointly optimizing its own problem-solving ability and collaboration, often resulting in a suboptimal balance between on-device and cloud resource utilization. We therefore pose the following two-fold question:

Figure 1: An illustration of our proposed RL-based unified training methodology and collaborative inference framework. (a) Training Framework: Two main scenarios where the lightweight on-device LLM learns to either solve problems independently or call for help. Note that the on-device LLM is trained offline before deployment on devices. (b) Collaborative Inference: The on-device LLM autonomously determines whether to process queries locally or invoke the cloud LLM.

*Can the on-device LLM be trained to autonomously decide when to invoke the cloud LLM, and can the acquisition of this routing capability be seamlessly integrated into the post-training stage alongside task-specific optimization?*

**Challenges.** Enabling on-device LLMs to autonomously decide when to invoke the cloud LLM during post-training could eliminate the need for an external router and simplify training procedures, yet achieving this capability presents several key challenges. First, on-device LLMs lack inherent routing capability even after task-specific post-training such as supervised fine-tuning (SFT) (Wei et al., 2021) or reinforcement learning from human feedback (RLHF) (Ziegler et al., 2019). Second, jointly optimizing routing with the model's own problem-solving ability introduces a coupled learning challenge, where the on-device LLM must simultaneously improve its own task performance while judiciously invoking cloud assistance. Third, collaborative inference is typically subject to practical limits on the offloading ratio, which makes it challenging to balance cloud calls with local computation. Overall, a comprehensive approach that jointly addresses task optimization, routing, and offloading constraints in a unified post-training framework remains elusive.

## 1.1 CONTRIBUTIONS

Motivated by these observations, we propose a reinforcement learning (RL)-based unified framework that enables the on-device LLM to both strengthen its own problem-solving ability and learn routing strategies, by integrating routing optimization directly into the post-training process (Figure 1). Concretely, we cast this as a reward maximization problem with a collaboration-aware hierarchical reward that assigns distinct reward signals to different responses, such as correct answers, wrong answers, and calls for help, while imposing a constraint on the usage of the cloud LLM to mitigate over-reliance. To solve this problem, we develop a group-adaptive policy gradient algorithm, characterized by (i) an unbiased group-based policy gradient estimator for stable optimization and (ii) adaptive prompt filtering to ensure complementary learning signals for both independent problem solving and calling for assistance.

Overall, we make the following contributions:

- *Unified formulation for problem solving and routing.* We formulate a reward maximization problem that integrates routing into the post-training process, enabling the on-device model to simultaneously enhance its own problem-solving ability and acquire routing strategies.

- *Group-adaptive policy gradient algorithm.* We propose a group-adaptive policy gradient algorithm, featured by an unbiased group-based policy gradient estimator and adaptive prompt filtering, to ensure stable optimization and provide complementary learning signals.

- *Extensive validation.* Through extensive experiments across diverse models and benchmarks, we demonstrated that our approach consistently outperforms baselines, maintains stable training, and significantly narrows the gap to full cloud LLM performance.

## 1.2 RELATED WORKS

**Reinforcement learning for large language models.** LLMs learn general language from large corpora (Radford et al., 2019; Brown et al., 2020), then are post-trained on task data to boost domain-specific performance (Ouyang et al., 2022). Post-training typically comprises SFT and RL-based tuning. Interest in the latter has surged following DeepSeek-R1-Zero (Guo et al., 2025), which omits SFT and relies solely on RL-based tuning. RL-based post-training was introduced for LLM alignment in (Ouyang et al., 2022), where Proximal Policy Optimization (PPO) is the primary algorithm. However, due to PPO's complexity and computational cost, simplified alternatives have been proposed, including Direct Preference Optimization (DPO) (Rafailov et al., 2023), ReMax (Li et al., 2024b), and Reinforce-Leave-One-Out (RLOO) (Ahmadian et al., 2024). Among these, Group Relative Policy Optimization (GRPO) (Shao et al., 2024) has gained particular traction for its simplicity and stability: it removes the learned critic and instead estimates baselines from group scores, reducing implementation complexity and variance while remaining competitive with PPO in performance. While RL-based post-training has undergone notable development, incorporating LLM coordination within this paradigm remains largely underexplored.

**Collaboration of large language models.** LLM collaboration aims to harness the complementary strengths of multiple models to enhance performance and efficiency. One line of work focuses on cascaded or ensemble-based routing across multiple LLMs. Lu et al. (2024) assumed LLMs have heterogeneous expertise and propose a reward-guided routing method that learns to send each query to the model most suited for it. Chen et al. (2023) designed a sequential LLM cascade where the models generate responses and confidence scores for each query sequentially, and the process halts once a response's score exceeds a preset threshold. Zhang et al. (2024) proposed a context-aware cascading policy that selects models under budget constraints. Another direction focuses on the collaboration of two LLMs, i.e., a weaker LLM and a stronger LLM. Specifically, Ding et al. (2024) fine-tuned a DeBERTa-v3-large model (He et al., 2020) to act as a router, which is anticipated to predict when the small model's output will match the large model's quality. Similarly, Ong et al. (2025) trained a router on human preference data to dispatch each query to either Mistral-8×7B or GPT-4. Notably, Mistral-8×7B, with 46.7B parameters, remains too large for device deployment. Despite these advances, most existing approaches rely on external routers or handcrafted policies, leaving the device model's intrinsic capacity for routing underutilized.

## 2 PROBLEM BACKGROUND

### 2.1 COLLABORATIVE FRAMEWORK FOR DEVICE-CLOUD LLMS

Consider a task with prompt set $\mathcal{D}$. The lightweight on-device LLM $\pi_\theta$, with tunable parameters $\theta$ designed for efficient deployment, may still struggle to handle certain prompts in $\mathcal{D}$ even after task-specific tuning, due to its limited capacity. To overcome this limitation, assistance from the cloud LLM $\pi_c$, which has a substantially broader knowledge scope, becomes essential. To maximize the potential of collaborative device-cloud LLMs on task $\mathcal{D}$, the most common solution is a two-stage pipeline (Ding et al., 2024; Ong et al., 2025; Yuan et al., 2025): (i) post-train the on-device model $\pi_\theta$, initialized from a pretrained model, to improve task-specific performance, and (ii) optimize a routing mechanism that decides, based on the relative capabilities of the on-device and cloud LLMs, whether each prompt should be handled locally or offloaded. We briefly review the most representative methods used for these two stages below.

**Stage I: RL-based post-training.** A leading technique for the first stage is GRPO. GRPO improves the on-device LLM by reinforcing relatively stronger responses while discouraging weaker ones. (Shao et al., 2024). Specifically, GRPO first samples a group of responses $\{\boldsymbol{y}_1, \boldsymbol{y}_2, \ldots, \boldsymbol{y}_G\}$ for each prompt $\boldsymbol{x}$ using the model. The rewards for these responses are denoted as $\{r_1, r_2, \ldots, r_G\}$. The normalized relative advantage of each response in the group is then computed as

$$A_i = (r_i - \text{mean}(\{r_i\}_{i=1}^G))/\text{std}(\{r_i\}_{i=1}^G). \tag{1}$$

The objective of GRPO is to optimize the model $\pi_\theta$ to maximize the expected relative advantage. The specific objective of GRPO is presented in Appendix A. While such post-training strengthens the model's own problem-solving ability, it does not endow the model with the ability to decide whether a prompt should be handled locally or offloaded, thereby necessitating a dedicated routing mechanism.

**Stage II: Routing optimization.** For the second stage, the conventional approach is to train an additional binary classifier, often implemented as another LLM, to make routing decisions. For each prompt, a response is sampled from the on-device and cloud models, and the router is trained to distinguish whether the on-device LLM can solve the prompt or if it should be offloaded to the cloud LLM. Formally, given a dataset of prompts $\mathcal{D}$, we assign binary labels $z \in \{0, 1\}$ indicating whether the on-device model suffices for prompt $\boldsymbol{x}$. The router is then trained by minimizing the binary cross-entropy loss: (Ding et al., 2024) $\mathcal{L}(\boldsymbol{w}) = -\frac{1}{|\mathcal{D}|} \sum_{\boldsymbol{x} \sim \mathcal{D}} \left( z \log p_w(\boldsymbol{x}) + (1 - z) \log(1 - p_w(\boldsymbol{x})) \right)$, where $p_w(\boldsymbol{x})$ denotes the router's predicted probability that the on-device LLM can handle $\boldsymbol{x}$.

## 2.2 LIMITATIONS

The two-stage pipeline suffers from several inherent drawbacks. First, the router is essentially a binary classifier. It is inherently difficult for such a classifier to judge whether the on-device LLM can solve a problem based solely on the prompt's surface pattern, since problems with similar structures may vary greatly in difficulty. Conversely, using a more powerful LLM with reasoning ability as the router would be inefficient and wasteful, since making a routing decision would require duplicating the reasoning process that should be performed by the on-device LLM. This redundancy adds unnecessary computation and storage overhead without contributing to solving the task. Finally, training and maintaining the router introduce additional engineering overhead, adding complexity to the system.

# 3 A UNIFIED TRAINING FRAMEWORK WITH REINFORCEMENT LEARNING

## 3.1 RL-BASED FORMULATION

To address these limitations, we introduce a unified perspective that embeds routing optimization into post-training, allowing the on-device LLM to improve its problem-solving ability while also learning routing strategies. Through fine-tuning parameters $\boldsymbol{\theta}$ of $\pi_\theta$, the on-device model not only strengthens its own problem-solving ability but also learns when to delegate to the cloud model $\pi_c$. Specifically, we anticipate the on-device LLM first attempts to generate a response locally and only invokes the cloud model $\pi_c$ at the end when it expects a better outcome. The resulting response $\boldsymbol{y}$ may be produced entirely by $\pi_\theta$ (i.e., $\boldsymbol{y} = \boldsymbol{y}^\theta$), or jointly with $\pi_c$ (i.e., $\boldsymbol{y} = [\boldsymbol{y}^\theta, \boldsymbol{y}^c]$).

To formalize this unified perspective, we cast training as a reward maximization problem, where the model seeks to optimize task performance subject to a budget on cloud model usage. This yields the following objective[1]:

$$\max_{\boldsymbol{\theta}} \; \mathbb{E}_{\boldsymbol{x} \sim \mathcal{D}} \left[ R(\boldsymbol{\theta}, \boldsymbol{x}) \right] := \mathbb{E}_{\boldsymbol{x} \sim \mathcal{D}} \mathbb{E}_{\boldsymbol{y}^\theta \sim \pi_\theta(\boldsymbol{x})} [r(\boldsymbol{x}, \boldsymbol{y})]$$
$$\text{subject to} \;\; \mathbb{E}[\mathbf{1}\{\boldsymbol{y} \sim (\pi_\theta, \pi_c)\}] \leq \rho \mathbb{E}[\mathbf{1}\{\boldsymbol{y} \sim \pi_\theta\}], \tag{2}$$

where $\mathcal{D}$ is the prompt set, $r(\boldsymbol{x}, \boldsymbol{y})$ denotes the reward assigned to response $\boldsymbol{y}$ under prompt $\boldsymbol{x}$, and $\mathbb{E}[\mathbf{1}\{\boldsymbol{y} \sim (\pi_\theta, \pi_c)\}]$ and $\mathbb{E}[\mathbf{1}\{\boldsymbol{y} \sim \pi_\theta\}]$ represent the frequencies of invoking the cloud LLM and generating the response solely with the on-device LLM, respectively. We assume the cloud model $\pi_c$ generates deterministically without adding stochasticity to training (Shi et al., 2024). The constraint in Problem (2), governed by $\rho$, restricts the usage ratio between cloud and on-device LLMs, thereby limiting over-reliance on the cloud LLM and encouraging the on-device LLM to solve problems independently. Notably, we compute the reward over the entire response to reflect overall response quality, but update the on-device LLM $\pi_\theta$ using its generated portion.

## 3.2 PROMPT AND REWARD DESIGN

The prompt and the reward function $r(\boldsymbol{x}, \boldsymbol{y})$ are two key components of Problem (2). In this subsection, we describe how we design the prompt template and reward to encourage the on-device model $\pi_\theta$ to invoke the cloud model $\pi_c$ when the task falls outside its capabilities.

**Prompt template.** As a next-token prediction model, an LLM tends to generate answers automatically, even when uncertain or incorrect. Without a dedicated prompt, it may fail to recognize its

---

[1]Note that on-device LLM is trained offline before deployment.

knowledge limitations and produce unreliable responses. To address this, we design a prompt template that guides the model to answer only when confident, and to invoke the cloud LLM for help when the question lies beyond its capabilities. The template is shown in Table 4 of Appendix D.

**Reward design.** Following the seminal work (Guo et al., 2025), we adopt a rule-based reward. To both foster effective coordination with the cloud LLM and maximize the on-device LLM's own problem-solving ability, we design a collaboration-aware hierarchical reward scheme comprising three components: *format*, *accuracy*, and *coordination* rewards, as detailed below.

- *Format reward:* This reward evaluates whether the on-device model $\pi_\theta$'s output follows the structure specified in the prompt. In particular, if the reasoning process is enclosed within `<think>` tags and the final answer is placed within `<answer>` tags, a format reward of $\alpha_f$ is assigned.

- *Accuracy reward:* This reward reflects the correctness of the response of the on-device model $\pi_\theta$. If the answer extracted from the on-device model $\pi_\theta$'s response is correct, an accuracy reward of $\alpha_a$ is assigned.

- *Coordination reward:* If the on-device model determines that it cannot solve the problem on its own and invokes the cloud LLM for assistance, a coordination reward of $\alpha_c$ is assigned, provided that the cloud LLM produces a correct answer.

We summarize all reward cases in (6) of Appendix D. In general, the reward weights satisfy $\alpha_a > \alpha_c > \alpha_f$, reflecting the priority of accuracy over coordination and format.

## 3.3 GRPO Fails to Coordinate On-Device and Cloud LLMs Collaboration

To jointly enhance task performance and collaboration, a natural approach is to optimize the on-device LLM $\pi_\theta$ using GRPO, a state-of-the-art post-training algorithm, together with the proposed collaboration-aware reward (Section 3.2).

**Initial investigation with GRPO.** We take Qwen2.5-3B-Instruct as the on-device model and DeepSeek-R1 as the cloud model. As the evaluation benchmark, we adopt the Countdown task (Pan et al., 2025), a mathematical puzzle in which players must combine a given set of numbers using the four basic arithmetic operations $(+, -, \times, \div)$ to reach a specific target number. For instance, given the numbers 75, 6, 2, and 3 with a target of 152, one valid solution is: $(75 \div 3) \times 6 + 2 = 152$. This task provides a wide range of problems with varying levels of difficulty, making it particularly suitable for the on-device and cloud LLMs collaboration settings. The reward parameters in Section 3.2 are set as $\alpha_a = 2$, $\alpha_c = 0.5$, and $\alpha_f = 0.2$. Figure 2 illustrates the evolution of the training reward and the call-for-cloud ratio with respect to training iterations.

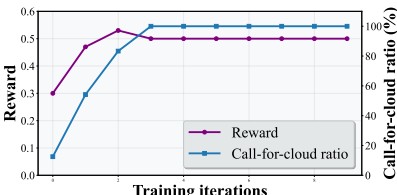

Figure 2: Rewards and call-for-cloud ratios versus training iterations. A naive approach offloads every prediction to the cloud server (i.e., a 100% call-for-cloud ratio), which violates the practical offloading constraints.

**Discussion.** As shown in Figure 2, the model trained with GRPO tends to converge to a low-quality suboptimal point that frequently calls for assistance from the cloud LLM. This behavior can be attributed to the following two factors:

- *Misalignment of normalized advantage with true rewards:* The normalized advantage used in GRPO, i.e., (1), can distort the actual value of different responses, which is problematic in our collaborative setting with hierarchical rewards. For instance, consider two groups of responses to the same prompt: Group A with rewards $[2.2, 0, 0, 0, 0, 0, 0, 0]$ and Group B with rewards $[0.5, 0, 0, 0, 0, 0, 0, 0]$, where 2.2 and 0.5 correspond to producing a correct answer independently and invoking the cloud model for assistance, respectively. After normalization, the advantages of the highest-reward responses in both groups are the same, despite the clear superiority of the 2.2 response. This causes the model to treat both responses as equally valuable, failing to recognize the greater merit of the independent correct answer.

- *Lack of penalty for over-reliance on the cloud LLM:* GRPO does not account for the constraint on invoking the cloud LLM. Since the assistance-invoking pattern is comparatively easy to learn, the model tends to overuse it, which leads to premature convergence to an undesirable policy.

---

**Algorithm 1** Collaborative Unified Training Framework with Group-Adaptive Policy Gradient

---

**Require:** Initial on-device LLM $\pi_\theta$ with parameters $\theta$, cloud LLM $\pi_c$, and prompt set $\mathcal{D}$

1: **for** iteration in $\{1, 2, \ldots, S\}$ **do**
2:      Sample a batch of prompts $\mathcal{D}_b$ from $\mathcal{D}$
3:      Sample $G$ responses $\{\boldsymbol{y}_1^\theta, \boldsymbol{y}_2^\theta, \ldots, \boldsymbol{y}_G^\theta\} \sim \pi_\theta(\cdot \mid \boldsymbol{x})$ for each prompt $\boldsymbol{x} \in \mathcal{D}_b$
4:      **for** each prompt $\boldsymbol{x} \in \mathcal{D}_b$ **do**
5:          **if** any response in $\{\boldsymbol{y}_1^\theta, \ldots, \boldsymbol{y}_G^\theta\}$ calls for help **then**
6:              Query cloud LLM $\pi_c$ to obtain $\boldsymbol{y}^c \sim \pi_c(\cdot \mid \boldsymbol{x})$         ▷ at most once for each prompt
7:              **for** each help-calling response $\boldsymbol{y}_i^\theta$ **do**
8:                  Set $\boldsymbol{y}_i \leftarrow [\boldsymbol{y}_i^\theta, \boldsymbol{y}^c]$         ▷ collaborative generation
9:              **end for**
10:          **else**
11:              Keep $\boldsymbol{y}_i \leftarrow \boldsymbol{y}_i^\theta$ for $i \in \{1, 2, \ldots, G\}$
12:          **end if**
13:          Evaluate rewards for responses $\{\boldsymbol{y}_i\}_{i=1}^G$ based on (6)         ▷ hierarchical reward
14:      **end for**
15:      Select prompts with both positive and negative responses, denoted as $\mathcal{D}_b^1$
16:      Select up to $\rho|\mathcal{D}_b^1|$ prompts for which none of the responses from $\pi_\theta$ are correct, but $\pi_c$ yields a correct answer, denoted as $\mathcal{D}_b^2$         ▷ adaptive prompt filtering
17:      Update $\theta \leftarrow \theta + \frac{\eta}{|\mathcal{D}_b^1 \cup \mathcal{D}_b^2|} \sum_{\boldsymbol{x} \in \mathcal{D}_b^1 \cup \mathcal{D}_b^2} \widehat{\nabla_\theta R}(\theta, \boldsymbol{x})$         ▷ group-adaptive policy gradient
18: **end for**

---

### 3.4 Proposed Group-Adaptive Policy Gradient Algorithm

To address the issues encountered by GRPO, we propose the Group-Adaptive Policy Gradient (GAPG) algorithm, tailored to the objective of Problem (2), which is featured by *group-level policy gradient*, designed to yield an unbiased gradient estimator of the optimization objective, and *adaptive prompt filtering*, developed to ensure complementary learning signals for both independent problem solving and calling for assistance. We introduced these two components in Sections 3.4.1 and 3.4.2, respectively, and summarize the overall procedures in Algorithm 1.

### 3.4.1 Group-level Policy Gradient

The reward function $r(\boldsymbol{x}, \boldsymbol{y})$ is non-differentiable with respect to the model parameters $\theta$ as it does not admit an analytic expression in terms of $\theta$. Consequently, standard gradient-based optimization algorithms cannot be directly applied. To address this challenge, we consider the expected reward defined in (2) and derive an unbiased gradient estimator. Different from conventional policy gradient methods (Barto, 2021), we introduce a group-level gradient estimator inspired by GRPO. Proposition 3.1 states the formulation, and its derivation, given in Appendix E, primarily employs the log-likelihood trick (Williams, 1992).

**Proposition 3.1 (Unbiased group gradient estimator)** *Given a prompt $\boldsymbol{x}$, draw a group of $G$ responses $\{\boldsymbol{y}_1, \ldots, \boldsymbol{y}_G\}$, where each response $\boldsymbol{y}_i$ may be produced entirely by the on-device policy $\pi_\theta$ (i.e. $\boldsymbol{y}_i = \boldsymbol{y}_i^\theta$) or jointly with the cloud policy $\pi_c$ (i.e. $\boldsymbol{y}_i = [\boldsymbol{y}_i^\theta, \boldsymbol{y}_i^c]$). Denote the reward for response $i$ as $r_i = r(\boldsymbol{x}, \boldsymbol{y}_i)$ and the group mean reward $\bar{r} = \frac{1}{G} \sum_{i=1}^G r_i$. For any $G \geq 2$, the following quantity*

$$\widehat{\nabla_\theta R}(\theta, \boldsymbol{x}) = \frac{G}{G-1} \frac{1}{G} \sum_{i=1}^G \nabla_\theta \log \pi_\theta(\boldsymbol{y}_i^\theta \mid \boldsymbol{x})(r_i - \bar{r}) \tag{3}$$

*is an* unbiased *estimator of the policy gradient* $\nabla_\theta R(\theta, \boldsymbol{x}) = \nabla_\theta \mathbb{E}_{\boldsymbol{y}^\theta \sim \pi_\theta(\boldsymbol{x})}[r(\boldsymbol{x}, \boldsymbol{y})]$.

With this gradient estimator, we can update the on-device LLM using stochastic gradient ascent:

$$\theta \leftarrow \theta + \eta \widehat{\nabla_\theta R}(\theta, \boldsymbol{x}), \quad \boldsymbol{x} \sim \mathcal{D},$$

iteratively, where $\eta$ is the learning rate. However, since this gradient is estimated from a single prompt, it suffers from high variance and does not account for the constraint on frequency of invoking the cloud LLM in Problem (2).

### 3.4.2 GROUP-ADAPTIVE PROMPT FILTERING

To further encourage the on-device LLM $\pi_\theta$ to explore and limit the frequency of invoking the cloud LLM $\pi_c$, we introduce an adaptive prompt filtering mechanism, guided by the constraint in Problem (2). The filtering mechanism focuses training on prompts that are most informative for learning the trade-off between independent problem solving and calling for external assistance. Specifically, in each training round, we sample $G$ responses for each prompt $x \in \mathcal{D}_b$ using models $[\pi_\theta, \pi_c]$. Based on the response quality, we form two prompt subsets:

- Set $\mathcal{D}_b^1$: Includes prompts where at least one of the $G$ sampled responses is generated correctly by the on-device model $\pi_\theta$. These prompts help the model learn to solve tasks on its own.

- Set $\mathcal{D}_b^2$: Includes up to $\rho|\mathcal{D}_b^1|$ prompts for which none of the sampled responses from on-device model $\pi_\theta$ are correct, but the cloud LLM $\pi_c$ provides a correct answer. These prompts are essential for guiding the model to identify situations in which leveraging the cloud LLM is beneficial.

By training $\pi_\theta$ on $\mathcal{D}_b^1 \cup \mathcal{D}_b^2$ (i.e., Step 17 of Algorithm 1), the on-device LLM receives complementary learning signals for both independent problem solving and calling for assistance.

## 4 EXPERIMENTS

**Datasets.** We fine-tune and test on-device LLMs on Countdown (Pan et al., 2025) and MATH-lighteval (Hendrycks et al., 2021). Additionally, we evaluate the models fine-tuned on the MATH-lighteval dataset against four widely used mathematical benchmarks: MATH-500 (Hendrycks et al., 2021), AMC23 (Lewkowycz et al., 2022), AIME24, and MinervaMath (Li et al., 2024a).

**On-device and cloud LLMs.** We employ Deepseek-R1 as the cloud model. For the Countdown task, we adopt Qwen2.5-3B-Instruct as the on-device model. For the MATH-lighteval task, we use on-device models of different sizes, Llama-3.2-1B-Instruct, Qwen2.5-1.5B-Instruct, and Llama-3.2-3B-Instruct. In Section 4.1, the call-for-cloud ratio is constrained to $30\%$ (i.e., $\rho/(1 + \rho)$) for all device-cloud collaboration scenarios, with requests exceeding this threshold redirected to the on-device LLM. Section 4.2 builds on this setting and further studies the impact of varying the ratio on performance.

**Baselines.** We compare our approach against the following baseline methods.

- Cloud LLM: All the queries are offloaded to the cloud model, i.e, Deepseek-R1 (Guo et al., 2025), which serves as a performance upper bound.

- Task-Tuning Only: Perform task-specific fine-tuning on the on-device model using GRPO (Shao et al., 2024). During inference, all predictions are made using the on-device LLM.

- Task-Tuning&Naive Offloading: The on-device LLM is first fine-tuned in the same way as in Task-Tuning Only, and then used in collaboration with the cloud LLM. During inference, a certain proportion of queries are randomly offloaded to the cloud LLM.

- Task-Tuning&Router: A two-stage approach where the on-device LLM is first fine-tuned as in Task-Tuning Only, and then an additional router (DeBERTa-v3-large) is trained to decide whether to call the cloud LLM based on the query and on-device LLM's response (Ding et al., 2024).

- Collaboration-Aware Tuning: The on-device LLM is fine-tuned with the GRPO algorithm, augmented by our proposed hierarchical reward (i.e., (6)) to encourage collaboration. Further details are provided in Section 3.3.

**Other Details.** All the experiments are conducted on a cluster equipped with 4 NVIDIA A100 GPUs, each with 80 GB of memory. The detailed hyperparameters are provided in Appendix F.

### 4.1 COUNTDOWN TASK: COMPARISON USING QWEN2.5-3B

Figure 3 compares the training reward and testing accuracy of our approach and the baselines on the Countdown task using the Qwen2.5-3B-Instruct model. Since Task-Tuning&Naive Offloading and Task-Tuning&Router employ the same RL process as Task-Tuning Only for tuning the

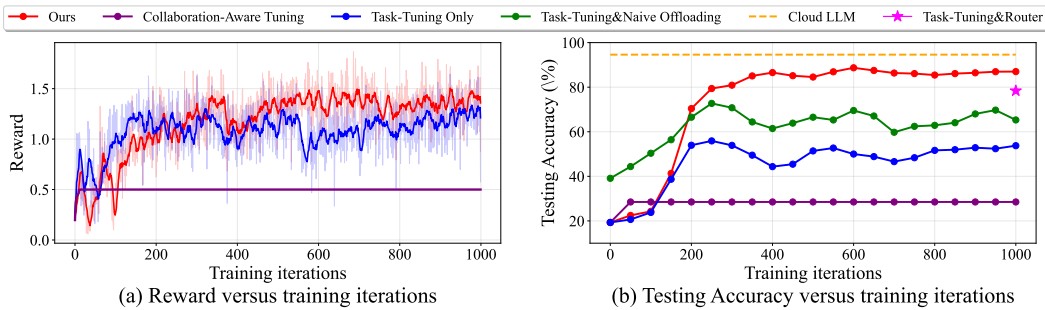

(a) Reward versus training iterations

(b) Testing Accuracy versus training iterations

Figure 3: Training reward and testing accuracy on the Countdown task with Qwen2.5-3B-Instruct. Our method consistently outperforms baselines, achieving higher rewards and accuracy.

on-device LLM, we report only the reward of Task-Tuning Only in Figure 3(a). As shown in Figure 3(a), the baseline Collaboration-Aware Reward converges to an always-call-for-cloud policy in this task, which is also discussed in Section 3.3. Additionally, both Task-Tuning Only and our proposed method exhibit similar reward growth in the early stages. However, after approximately 200 steps, our approach consistently outperforms the baseline and maintains a clear margin as training progresses. Turning to Figure 3(b), we compare the testing accuracy, which further confirms the consistent superiority of our approach over the baselines.

Figure 3(b) compares the testing accuracy versus the baselines. Note that the Router is trained only after completing task-specific training; therefore, we report only the final accuracy of Task-Tuning&Router in Figure 3(b). As shown in Figure 3(b), our method achieves performance approaching that of the Cloud LLM and surpasses all baselines. In particular, it improves accuracy by approximately 30%, nearly matching the cloud offloading rate, relative to Task-Tuning Only. This demonstrates that our approach equips the on-device model with coordination ability without compromising its problem-solving ability, highlighting the effectiveness of our collaborative unified training methodology.

## 4.2 MATH TASK: COMPARISON ACROSS QWEN2.5-1.5B AND LLAMA-3.2 MODELS

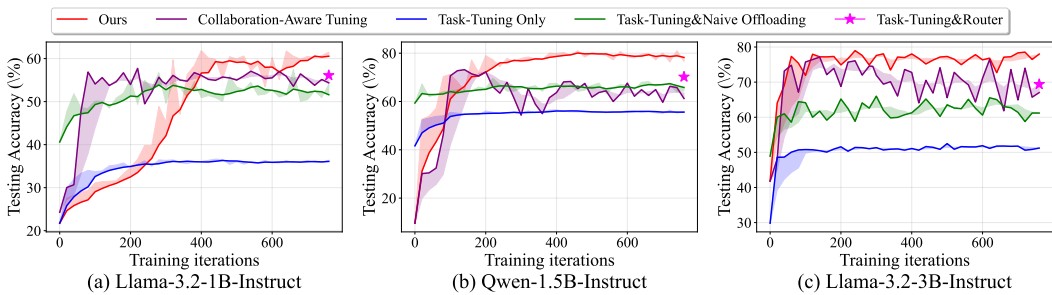

(a) Llama-3.2-1B-Instruct

(b) Qwen-1.5B-Instruct

(c) Llama-3.2-3B-Instruct

Figure 4: Testing accuracy versus training iterations on the MATH-lighteval dataset. Our method consistently outperforms baselines across three on-device models, while also exhibiting stable training behavior, demonstrating its effectiveness and robustness.

**Training dynamics.** We further evaluate our approach on the MATH-lighteval dataset using three on-device models of varying sizes (1B, 1.5B, and 3B), with the testing accuracy curves across training iterations shown in Figure 4. Unlike on Countdown, Collaboration-Aware Tuning does not collapse on MATH-lighteval, likely because the initial models possess stronger prior knowledge of this widely used benchmark, which makes them less vulnerable even under biased fine-tuning. While our method initially lags behind some baselines in the early training phase, it consistently surpasses them as training progresses and ultimately achieves the highest accuracy across all model sizes. By contrast, Task-Tuning&Router exhibits noticeably worse performance, as the router fails to provide effective offloading decisions. A potential reason is that problems in this task are often

Table 1: Accuracy (%) of our approach and baselines, tuned on MATH-lighteval and evaluated on five math benchmarks. Our method achieves the highest average accuracy across both models.

| Model | Method | MATH-lighteval | MATH-500 | AMC23 | AIME24 | MinervaMath | Avg. |
|---|---|---|---|---|---|---|---|
| | Cloud LLM | 98.4 | 97.3 | 97.5 | 79.8 | 80.9 | 90.8 |
| Qwen2.5-1.5B-Instruct | Task-Tuning Only | 56.1 | 54.8 | 35.0 | 0.0 | 20.6 | 33.3 |
| | Task-Tuning&Naive Offloading | 67.2 | 67.4 | 50.0 | 23.3 | 38.2 | 49.2 |
| | Collaboration-Aware Tuning | 61.5 | 61.2 | 35.0 | 23.3 | 33.5 | 42.9 |
| | Task-Tuning&Router | 70.9 | 72.2 | 55.0 | 23.3 | 36.8 | 51.6 |
| | **Ours** | **78.0** | **81.6** | **57.5** | **23.3** | **40.8** | **56.2** |
| Llama-3.2-3B-Instruct | Task-Tuning Only | 51.2 | 43.0 | 27.5 | 10.0 | 19.1 | 30.2 |
| | Task-Tuning&Naive Offloading | 65.1 | 59.0 | 45.0 | 30.0 | 37.1 | 47.2 |
| | Collaboration-Aware Tuning | 66.8 | 59.6 | 42.5 | 30.0 | 36.8 | 47.1 |
| | Task-Tuning&Router | 69.4 | 61.0 | 45.0 | 30.0 | 37.1 | 48.5 |
| | **Ours** | **79.5** | **68.6** | **52.5** | **33.3** | **43.4** | **55.5** |

structurally similar, and the prompt alone does not reliably indicate their difficulty. Additionally, the slower early-stage convergence of our approach arises because our methodology explicitly balances on-device LLM's own problem-solving ability with cloud LLM coordination. Nevertheless, as Figure 4 demonstrates, this joint optimization yields clear long-term benefits: our approach converges to substantially higher accuracy, underscoring its effectiveness.

**Evaluation outside the training task.** Beyond MATH-lighteval, we report the performance of the tuned Qwen2.5-1.5B and Llama-3.2-3B models on four additional widely used mathematical benchmarks in Table 1. As shown, our method achieves the highest accuracy across all benchmarks, with average improvements between 6 and 8 points compared to baselines, demonstrating that the coordination ability acquired through our approach generalizes well to new tasks. We also observe that while baselines Collaboration-Aware Tuning or Task-Tuning&Router sometimes achieve competitive performance on individual datasets, they fail to maintain consistent accuracy across all benchmarks. In contrast, our approach yields balanced improvements across both easier (MATH-lighteval, MATH-500) and more challenging datasets (AMC23, AIME24, MinervaMath).

**Impact of call-for-cloud ratio.** In Figure 5, we evaluate the impact of the call-for-cloud ratio on the testing accuracy. For the Collaboration-Aware Tuning baseline, the trained on-device model converges to a call-for-cloud ratio of approximately 59.7%. To evaluate it at lower target ratios (e.g., 20% or 40%), we truncate additional calls once the target ratio is reached. As shown in Figure 5, our method achieves the strongest performance across all ratios, delivering notable gains even with moderate cloud reliance (20-40%) and nearly matching the Cloud LLM at 60%. In contrast, Collaboration-Aware Tuning suffers performance degradation at low ratios (e.g., 20%), as it fails to effectively balance the development of coordination with strengthening problem-

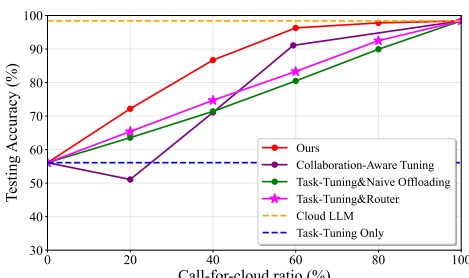

Figure 5: Impact of call-for-cloud ratio on accuracy. Our approach rapidly narrows the gap to Cloud LLM as the ratio increases

solving. Meanwhile, the performance of Task-Tuning&Naive Offloading and Task-Tuning&Router improves steadily but consistently remains inferior. Overall, the proposed collaborative unified training consistently outperforms separate routing and naive offloading.

## 5 CONCLUSION

We proposed a collaborative device-cloud LLM inference framework where the on-device LLM itself decides whether to invoke the cloud LLM at the end of its solving process. To endow this capability, we formulated a reward maximization problem that integrates routing optimization into post-training, enabling the on-device LLM to strengthen its problem-solving ability while developing coordination with the cloud LLM. To solve this problem, we developed a group-adaptive policy gradient algorithm with a group-level policy gradient for unbiased optimization and adaptive prompt filtering to provide complementary learning signals. Through extensive experiments across diverse models and benchmarks, we demonstrated that our approach consistently outperforms baselines, maintains stable training, and significantly narrows the gap to full cloud LLM performance.

## LIMITATIONS

Our evaluation primarily focuses on mathematical reasoning tasks where correctness-based rewards are readily available. Extending our framework to more open-ended tasks would require designing reliable quality-aware rewards. Achieving this may necessitate incorporating LLM-as-a-Judge mechanisms to provide more nuanced feedback. Advancing this aspect represents an important direction for future research.

## REPRODUCIBILITY STATEMENT

This paper provides all the necessary information to reproduce the main experimental results. The datasets used are publicly available, while the model architectures, hyperparameters, and training details are documented in Section 4. The full implementation code is included in the supplementary material of our submission.

## LLM USAGE

We used ChatGPT (GPT-5), as an assistive tool only for improving the clarity and readability of the manuscript. The LLM was employed to polish grammar and rephrase sentences for conciseness. It was not used for research ideation, methodological design, and experimental implementation. All scientific content, including problem formulation, algorithm development, analysis, and experiments, was entirely conceived and executed by the authors.

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

## A  Training Objective of GRPO

The trajectory-level normalized relative advantage defined in (1) is assigned to each token, i.e., $A_{i,t} = A_i$. Accordingly, the training objective of GRPO is given by

$$
\max_{\boldsymbol{\theta}} \mathbb{E}_{\boldsymbol{x} \sim \mathcal{D}, \{\boldsymbol{y}_i\}_{i=1}^G \sim \pi_{\theta_{\text{old}}}(\boldsymbol{x})} \frac{1}{G} \sum_{i=1}^{G} \frac{1}{|\boldsymbol{y}_i|} \sum_{t=1}^{|\boldsymbol{y}_i|} \left\{ \min \left[ \frac{\pi_\theta \left( y_{i,t} \mid \boldsymbol{x}, \boldsymbol{y}_{i,<t} \right)}{\pi_{\theta_{\text{old}}} \left( y_{i,t} \mid \boldsymbol{x}, \boldsymbol{y}_{i,<t} \right)} A_{i,t}, \right. \right.
$$

$$
\left. \text{clip} \left( \frac{\pi_\theta \left( y_{i,t} \mid \boldsymbol{x}, \boldsymbol{y}_{i,<t} \right)}{\pi_{\theta_{\text{old}}} \left( y_{i,t} \mid \boldsymbol{x}, \boldsymbol{y}_{i,<t} \right)}, 1 - \varepsilon, 1 + \varepsilon \right) A_{i,t} \right] - \beta \mathbb{D}_{\text{KL}} \left[ \pi_\theta || \pi_{\text{ref}} \right] \right\}, \tag{4}
$$

$$
\text{where } A_i = (r_i - \text{mean}(\{r_i\}_{i=1}^G)) / \text{std}(\{r_i\}_{i=1}^G).
$$

In particular, $\pi_{\theta_{\text{old}}}$ denotes the stale policy used for sampling, and $\pi_{\text{ref}}$ is the reference model, typically set to the initial policy to penalize deviations from the starting point. The hyperparameters $\epsilon$ and $\beta$ control the clipping range and the strength of KL regularization, respectively.

## B  Discussion on Dr. GRPO

Prior to our work, Liu et al. (2025) also proposed to remove the standard deviation from the advantage computation optimize the following objective

$$
\max_{\boldsymbol{\theta}} \mathbb{E}_{\boldsymbol{x} \sim \mathcal{D}} \frac{1}{G} \sum_{i=1}^{G} \sum_{t=1}^{|\boldsymbol{y}_i|} \left\{ \min \left[ \frac{\pi_\theta \left( y_{i,t} \mid \boldsymbol{x}, \boldsymbol{y}_{i,<t} \right)}{\pi_{\theta_{\text{old}}} \left( y_{i,t} \mid \boldsymbol{x}, \boldsymbol{y}_{i,<t} \right)} A_{i,t}, \text{clip} \left( \frac{\pi_\theta \left( y_{i,t} \mid \boldsymbol{x}, \boldsymbol{y}_{i,<t} \right)}{\pi_{\theta_{\text{old}}} \left( y_{i,t} \mid \boldsymbol{x}, \boldsymbol{y}_{i,<t} \right)}, 1 - \varepsilon, 1 + \varepsilon \right) A_{i,t} \right] \right\},
$$

$$
\text{where } A_i = (r_i - \text{mean}(\{r_i\}_{i=1}^G)). \tag{5}
$$

However, our work is still different from Dr. GRPO. First, the key components of our algorithm include the hierarchical reward design, adaptive prompt filtering, and the removal of standard-deviation normalization. Our motivation for identifying the issue of standard-deviation normalization in the GRPO primarily stems from the hierarchical reward formulation, which, together with adaptive prompt filtering, is unique to our method. Second, Dr. GRPO follows the GRPO pipeline and modifies its advantage calculation, whereas ours is more like a policy gradient method. Consequently, our algorithm does not rely on clipping or importance sampling, both of which are inherited by GRPO-style algorithm. The removal of standard-deviation normalization arises naturally from the policy-gradient derivation. Third, and more importantly, the core contribution of our work is not a new stand-alone RL algorithm. The primary novelty lies in formulating and solving the on-device cloud LLM collaboration problem in an end-to-end manner, which, to the best of our knowledge, has not been explored in prior literature. Our contributions are three-fold:

- **Collaboration paradigm.** We propose the first principled approach to integrate routing behavior directly into the on-device LLM through RL-based post-training, enabling the model to learn when to rely on its own reasoning and when to offload to the cloud LLM.

- **Problem formulation.** We introduce a hierarchical reward design that jointly captures task correctness and device-cloud coordination. This formulation is unique to the collaborative inference setting and is essential for this methodology.

- **Algorithmic design.** We develop a policy-gradient method tailored to this formulation, featuring group-response sampling and adaptive prompt filtering.

## C  Ablation Study

To validate the contribution of each component in our framework, we conduct ablation studies on the hierarchical reward weights and the group size used in the group-based gradient estimator. These experiments aim to show that both components are indispensable.

### C.1  Hierarchical Reward Weights

As the formatting reward $\alpha_f$ is generally smaller than the coordination reward $\alpha_c$. Therefore, the dominant factor governing the trade-off between independent solving and cloud assistance is the

relative magnitude of accuracy reward $\alpha_a$ versus coordination reward $\alpha_c$. We report the call-for-help ratios and overal accuracy obtained under different combinations of $\alpha_a$ and $\alpha_c$. The following results are based on the Qwen2.5-1.5B-Instruct model evaluated on MATH-lighteval. The formatting reward is fixed at $\alpha_f = 0$. Note that the accuracies shown below do not account for the call-for-help budget constraint.

Table 2: Effect of hierarchical reward weights on call-for-help behavior and accuracy.

| Reward weights | Call-for-help ratio | Local accuracy | Overall accuracy |
|---|---|---|---|
| $\alpha_a = 1, \alpha_c = 0$ | 0 | 56.1 | 56.1 |
| $\alpha_a = 1, \alpha_c = 0.3$ | 29.7 | 48.9 | 78.0 |
| $\alpha_a = 1, \alpha_c = 0.35$ | 42.1 | 47.4 | 88.7 |
| $\alpha_a = 1, \alpha_c = 0.5$ | 50.3 | 43.1 | 92.3 |

As we can see from Table 2, different combinations of $\alpha_a$ and $\alpha_c$ lead to different call-for-help ratios. This demonstrates that the hierarchical reward shapes collaboration behavior and enables controllable call-for-help ratios.

### C.2 THE GROUP SIZE OF THE GROUP ESTIMATOR

We next analyze the role of the group size $G$ used for group response sampling in the gradient estimator. This component is inspired by GRPO. We train the Qwen2.5-1.5B-Instruct model with group sizes $G = \{4, 8, 16, 32\}$ and then evaluate it over MATH tasks.

Table 3: Overall accuracy of Qwen2.5-1.5B-Instruct on MATH tasks under different group sizes.

| Group size | MATH-lighteval | MATH-500 | AMC23 | AIME24 | MinervaMath | Avg. |
|---|---|---|---|---|---|---|
| 4 | 56.0 | 61.3 | 35.0 | 23.3 | 22.1 | 39.5 |
| 8 | 78.0 | 81.6 | 57.5 | 23.3 | 40.8 | 56.2 |
| 16 | 78.2 | 81.1 | 57.5 | 23.3 | 41.2 | 56.3 |
| 32 | 78.3 | 81.9 | 60.0 | 23.3 | 41.5 | 57.0 |

As shown in Table 3, using a small group size, such as $G = 4$, leads to a substantial drop in accuracy across all benchmarks, while increasing $G$ to 8 or 16 largely stabilizes performance, indicating that sufficient response diversity is essential for the training. The group estimator stabilizes training via diverse group sampling and improves accuracy. Additionally, increasing to $G = 32$ provides marginal improvement but substantially higher cost, indicating that moderate group size offers the best trade-off.

Overall, with the empirical results shown above, we see that the framework does not function properly when either component is removed or weakened, which validates that both the hierarchical reward and the group-based gradient estimator are essential for enabling effective device-cloud collaboration.

## D PROMPT TEMPLATE AND REWARD DETAILS

**Prompt.** We design two prompt templates that explicitly instruct the model to answer only when confident, as shown in Table 4. Template II provides stricter guidance than Template I and can be applied when the model fails to recognize its knowledge limitations under Template I.

In our experiments, we use Template I for the Countdown task and Template II for the MATH-lighteval task. The tunable hyperparameter *number* is set to 6.

**Reward.** All possible rewards discussed in Section 3.2 are summarized in the following equation, which integrates the three components of our collaboration-aware reward design, *format*, *accuracy*, and *coordination*, to jointly enforce structural correctness, problem-solving accuracy, and effective

Table 4: Prompt templates for training the on-device LLM in the collaborative device–cloud framework. The placeholder *question* and *number* will be replaced with the actual question and an appropriate number during training.

---

**Template I.** *System prompt:* You are a helpful assistant. You first think about the reasoning process in your mind and then provide the user with the answer. Show all your reasoning in `<think> </think>` tags. And return the final answer in `<answer> </answer>` tags at the end. If you did not find a solution after a thorough reasoning process, you can ask for external assistance at the end, for example, `<unknown>` I need external assistance `</unknown>`.
User: *question*. Let me solve this step by step.

---

**Template II.** *System prompt:* You are given a math problem. Solve it step by step. Organize your thoughts using this format: Step 1: ..., Step 2: ..., Step 3: ..., and so on. Put your final answer within `\boxed{}`. If you cannot solve the problem after *number* reasoning steps, stop reasoning and return: `<unknown>` I need external assistance `</unknown>`.
User: *question*. Let's think step by step.

---

device–cloud collaboration:

$$r(\boldsymbol{x}, \boldsymbol{y}) = \begin{cases} \alpha_a + \alpha_f, & \text{if } \boldsymbol{y} = \boldsymbol{y}^\theta \text{ contains the correct answer and follows the required format} \\ \alpha_a, & \text{if } \boldsymbol{y} = \boldsymbol{y}^\theta \text{ contains the correct answer but violates the format} \\ \alpha_c, & \text{if } \boldsymbol{y} = [\boldsymbol{y}^\theta, \boldsymbol{y}^c] \text{ contains the correct answer} \\ \alpha_f, & \text{if } \boldsymbol{y} = \boldsymbol{y}^\theta \text{ does not contain the correct answer but follows the format} \\ 0, & \text{otherwise.} \end{cases} \quad (6)$$

## E  PROOF OF PROPOSITION 3.1

Recall the estimator

$$\widehat{\nabla_\theta R}(\boldsymbol{\theta}, \boldsymbol{x}) = \frac{G}{G-1} \frac{1}{G} \sum_{i=1}^{G} \left[ \nabla_\theta \log \pi_\theta(\boldsymbol{y}_i^\theta \mid \boldsymbol{x}) \right] (r_i - \bar{r}), \quad \bar{r} = \frac{1}{G} \sum_{j=1}^{G} r_j.$$

Because each $\boldsymbol{y}_i^\theta \sim \pi_\theta(\boldsymbol{x})$ and $\int \pi_\theta(\boldsymbol{y} \mid \boldsymbol{x}) \, d\boldsymbol{y} = 1$, the following identity holds for every $i$:

$$\begin{aligned} \mathbb{E}_{\boldsymbol{y}_i^\theta \sim \pi_\theta(\boldsymbol{x})} \left[ \nabla_\theta \log \pi_\theta(\boldsymbol{y}_i^\theta \mid \boldsymbol{x}) \right] &= \int \pi_\theta(\boldsymbol{y} \mid \boldsymbol{x}) \nabla_\theta \log \pi_\theta(\boldsymbol{y} \mid \boldsymbol{x}) \, d\boldsymbol{y} \\ &= \int \nabla_\theta \pi_\theta(\boldsymbol{y} \mid \boldsymbol{x}) \, d\boldsymbol{y} \\ &= \nabla_\theta \int \pi_\theta(\boldsymbol{y} \mid \boldsymbol{x}) \, d\boldsymbol{y} \\ &= 0. \end{aligned} \quad (7)$$

Following the log-likelihood trick, we have

$$\begin{aligned} \mathbb{E}_{\boldsymbol{y}_i^\theta \sim \pi_\theta(\boldsymbol{x})} \left[ \nabla_\theta \log \pi_\theta(\boldsymbol{y}_i^\theta \mid \boldsymbol{x}) \, r(\boldsymbol{x}, \boldsymbol{y}_i) \right] &= \int \pi_\theta(\boldsymbol{y}_i^\theta \mid \boldsymbol{x}) \nabla_\theta \log \pi_\theta(\boldsymbol{y}^\theta \mid \boldsymbol{x}) \, r(\boldsymbol{x}, \boldsymbol{y}) d\boldsymbol{y} \\ &= \int \nabla_\theta \pi_\theta(\boldsymbol{y}^\theta \mid \boldsymbol{x}) \, r(\boldsymbol{x}, \boldsymbol{y}) d\boldsymbol{y} \\ &= \nabla_\theta \int \pi_\theta(\boldsymbol{y}^\theta \mid \boldsymbol{x}) \, r(\boldsymbol{x}, \boldsymbol{y}) d\boldsymbol{y} \\ &= \nabla_\theta \mathbb{E}_{\boldsymbol{y}^\theta \sim \pi_\theta(\boldsymbol{x})} \left[ r(\boldsymbol{x}, \boldsymbol{y}) \right] \\ &= \nabla_\theta R(\boldsymbol{\theta}, \boldsymbol{x}). \end{aligned} \quad (8)$$

Using linearity of expectation, we have

$$\mathbb{E}[\widehat{\nabla_\theta R}(\boldsymbol{\theta}, \boldsymbol{x})] = \frac{G}{G-1} \frac{1}{G} \sum_{i=1}^{G} \mathbb{E} \left[ \nabla_\theta \log \pi_\theta(\boldsymbol{y}_i^\theta \mid \boldsymbol{x}) \, (r_i - \bar{r}) \right].$$

Because the $G$ terms are identically distributed, replace the sum by a single expectation:

$$
\begin{aligned}
\mathbb{E}\big[\widehat{\nabla_\theta R}(\boldsymbol{\theta}, \boldsymbol{x})\big] =& \frac{G}{G-1}\, \mathbb{E}\big[\nabla_\theta \log \pi_\theta(\boldsymbol{y}_1^\theta \mid \boldsymbol{x})\,(r_1 - \bar{r})\big] \\
=& \frac{G}{G-1}\, \mathbb{E}\left[\nabla_\theta \log \pi_\theta(\boldsymbol{y}_1^\theta \mid \boldsymbol{x})\left(\big(1 - \tfrac{1}{G}\big)r_1 - \tfrac{1}{G}\sum_{j\neq 1} r_j\right)\right] \\
=& \mathbb{E}\big[\nabla_\theta \log \pi_\theta(\boldsymbol{y}_1^\theta \mid \boldsymbol{x})r_1\big] + \frac{1}{G-1}\sum_{j\neq 1}\mathbb{E}\big[\nabla_\theta \log \pi_\theta(\boldsymbol{y}_1^\theta \mid \boldsymbol{x})r_j\big].
\end{aligned}
\tag{9}
$$

For the term with $r_1$, following (8), we have
$$
\mathbb{E}\big[\nabla_\theta \log \pi_\theta(\boldsymbol{y}_1^\theta \mid \boldsymbol{x})\,r_1\big] = \nabla_\theta R(\boldsymbol{\theta}, \boldsymbol{x}).
\tag{10}
$$

For the term with $r_j, j \neq 1$, the random variables $\nabla_\theta \log \pi_\theta(\boldsymbol{y}_1^\theta \mid \boldsymbol{x})$ and $r_j$ are independent, giving
$$
\begin{aligned}
\mathbb{E}\big[\nabla_\theta \log \pi_\theta(\boldsymbol{y}_1^\theta \mid \boldsymbol{x})\,r_j\big] &= r_j \cdot \mathbb{E}\big[\nabla_\theta \log \pi_\theta(\boldsymbol{y}_1^\theta \mid \boldsymbol{x})\big] \\
&= 0,
\end{aligned}
\tag{11}
$$

where the second inequality comes from (7). Plugging (10) and (11) into (9) gives rise to
$$
\mathbb{E}\big[\widehat{\nabla_\theta R}(\boldsymbol{\theta}, \boldsymbol{x})\big] = \nabla_\theta R(\boldsymbol{\theta}, \boldsymbol{x}).
$$

This completes the proof.

## F    DETAILS ON HYPERPARAMETERS AND DATASETS

**Details on Hyperparameters.** Unless stated otherwise, the hyperparameters used for the Countdown task under the Qwen2.5-3B-Instruct and MATH-lighteval task under Llama-3.2-1B-Instruct, Qwen2.5-1.5B-Instruct, and Llama-3.2-3B-Instruct are as follows.

Table 5: The hyperparameters for the Countdown task under the Qwen2.5-3B-Instruct model and the MATH-lighteval task under Llama-3.2-1B-Instruct, Qwen2.5-1.5B-Instruct, and Llama-3.2-3B-Instruct models.

| Hyperparameter | Countdown & Qwen | Math-lighteval & LLaMA-3.2-1.5B/Qwen-2.5-1B/LLaMA-3.2-3B |
|---|---|---|
| Batch size $\|\mathcal{D}_b\|$ | 32 | 128 |
| Group size $G$ | 8 | 8 |
| Max prompt length | 256 | 1024 |
| Max response length | 720 | 1024 |
| Learning rate $\eta$ | 5e-6 | 2e-6 |
| Total training steps $S$ | 1000 | 1160/780/780 |
| Sampling temperature for training | 1.0 | 1.0 |
| Sampling temperature for evaluation | 0 | 0 |
| Format reward | 0.2 | 0 |
| Accuracy reward | 2 | 1 |
| Coordination reward | 0.5 | 0.2/0.3/0.6 |
| Cloud-device usage ratio $\rho$ | 3/7 | 3/7 |

**Details on Datasets.** We fine-tune the on-device LLM on two datasets, Countdown (Pan et al., 2025) and MATH-lighteval (Hendrycks et al., 2021), respectively. The Countdown task is an arithmetic puzzle where the model must combine a given set of numbers using basic arithmetic operations $(+, -, \times, \div)$ to reach a specified target number. We randomly select 6,000 problems from the Countdown dataset, using 5,000 for training and 1,000 for testing. The MATH-lighteval dataset comprises 12,500 problems drawn from mathematics competitions, covering topics such as algebra, geometry, counting and probability, number theory, and precalculus. We use 7,500 samples for training and 5,000 for testing. Additionally, we evaluate the models fine-tuned on the MATH-lighteval dataset against four widely used mathematical benchmarks: MATH-500 (Hendrycks et al., 2021), AMC23 (Lewkowycz et al., 2022), AIME24, and MinervaMath (Li et al., 2024a).

# G  Learning Dynamics of the Router in the Two-Stage Baseline

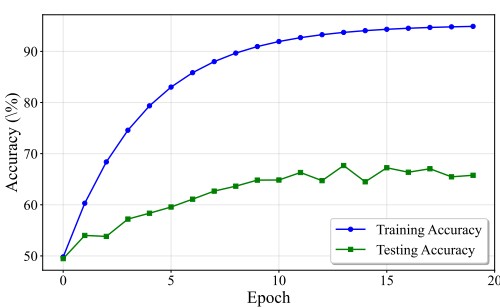

Figure 6: Training dynamics of the router, averaged over two routers

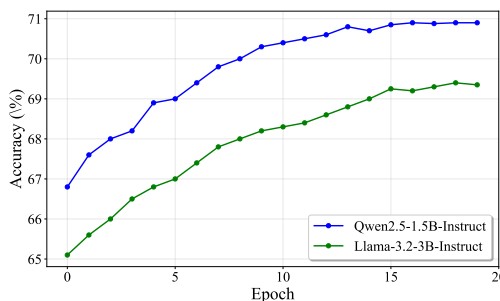

Figure 7: Performance of the two-stage baseline versus the training epochs of the router under a 30% call-for-help budget

This section examines the training trajectory of the router module under the two-stage approach baseline, i.e., Task-Tuning&Router. We report the accuracy curve over router-training epochs to illustrate its convergence behavior and confirm that the baseline was fully optimized before comparison. The results in Figure 6 are averaged over two routers trained on Llama-3.2-3B-Instruct and Qwen2.5-1.5B-Instruct. As we can see, the router achieves high training accuracy and exhibits clear convergence, confirming that the two-stage baseline was fully optimized. However, the testing accuracy remains lower, indicating poor generalization. Although the router can fit the training distribution well, its predictions on unseen data remain unreliable because surface patterns of the queries do not faithfully reflect the true difficulty of the underlying tasks. As shown in Figure 7, the router's task performance improves as training progresses. Nonetheless, the improvement is limited, further reflecting the inherent difficulty of predicting problem difficulty from surface-level query features alone.

# H  Number of Cloud Calls Comparison

During training, the cloud LLM is invoked at most once per prompt. In our implementation, we cache the cloud LLM's response the first time a prompt triggers a cloud call and reuse this cached result for all subsequent training iterations. Therefore, each prompt incurs at most a single cloud call, and the overall training cost is bounded by the size of the training set.

Table 6 reports the actual number of cloud calls incurred in training for Figures 3 and 4. The baselines Task-Tuning Only and Task-Tuning & Naive Offloading do not use the cloud LLM and thus have zero cost, which are thus omitted from this table.

| Methods | Ours | Task-Tuning & Router | Collaboration-Aware Tuning |
|---------|------|----------------------|----------------------------|
| Figure 3 | 2519 | 2540 | 6000 |
| Figure 4 | 2883 | 3075 | 6746 |

Table 6: Number of cloud calls incurred during training for the models used in Figures 3 and 4.

As shown in the table, among the baselines that invoke the cloud LLM during training, our method does not incur a higher cloud-call cost.

# I  Can the On-Device LLM Exit Early Before Calling for Help?

This section presents a case study demonstrating that the on-device LLM does not always complete a full reasoning trace before routing. Instead, the model sometimes halts early when it detects low confidence and calls for help.

## Two representative call-for-help patterns

### Long reasoning before calling for help

**Question:** Two positive numbers $p, q$ satisfy

$$p + q = pq, \qquad p - q = 7.$$

Compute

$$\frac{1}{\frac{1}{p^2} + \frac{1}{q^2}}$$

in the form $\frac{a + b\sqrt{c}}{d}$, where $a, b$ do not both share a factor with $d$ and $c$ is squarefree. Find $a + b + c + d$.

**Response: Step 1.** Let's denote the two numbers as $p$ and $q$. We are given two conditions:

$$p + q = pq$$

$$p - q = 7$$

**Step 2.** From the first condition, we can rearrange to get:

$$pq - p - q = 0$$

Adding 1 to both sides:

$$pq - p - q + 1 = 1$$

This can be factored as:

$$(p - 1)(q - 1) = 1$$

**Step 3.** Now we have the system:

$$p + q = pq, \qquad p - q = 7$$

From the second equation, solve for $p$:

$$p = q + 7$$

Substituting into $p + q = pq$,

$$(q + 7) + q = (q + 7)q$$

Simplifying:

$$2q + 7 = q^2 + 7q$$

Rearranging:

$$q^2 + 5q - 7 = 0$$

Using the quadratic formula,

$$q = \frac{-5 \pm \sqrt{25 + 28}}{2}$$

$$q = \frac{-5 \pm \sqrt{53}}{2}$$

Since $q > 0$,

$$q = \frac{-5 + \sqrt{53}}{2}$$

Substitute into $p = q + 7$:

$$p = \frac{-5 + \sqrt{53}}{2} + 7 = \frac{-5 + \sqrt{53} + 14}{2} = \frac{9 + \sqrt{53}}{2}$$

**Step 4.** Now compute:

$$\frac{1}{\frac{1}{p^2} + \frac{1}{q^2}} = \frac{p^2 q^2}{p^2 + q^2}$$

We know $p + q = pq$, so:

$$p^2 + q^2 = (p + q)^2 - 2pq$$

From $p + q = pq$,

$$p^2 + q^2 = p^2 + q^2$$

Now compute:

$$p^2 q^2 = (pq)^2 = p^2 q^2$$

$$p^2 + q^2 = (p + q)^2 - 2pq = p^2 + q^2$$

Thus:

$$\frac{1}{\frac{1}{p^2} + \frac{1}{q^2}} = \frac{p^2 q^2}{p^2 + q^2} = \frac{p^2 q^2}{p^2 + q^2} = 1$$

**I need external assistance.**

---

### A short reasoning before calling for help (Early exit)

**Question:** For what real values of $x$ is $-4 < x^4 + 4x^2 < 21$ satisfied? Express your answer in interval notation.
**Response:** **Step 1:** We need to solve the inequality

$$-4 < x^4 + 4x^2 < 21.$$

**Step 2:** Let's start by solving the left part of the inequality,

$$-4 < x^4 + 4x^2.$$

**Step 3:** Let's solve the right part of the inequality,

$$x^4 + 4x^2 < 21.$$

I need external assistance.
**Question:** The rightmost non-zero digit in $(1001001)(1010101) + (989899)(1001001) - (1001)(989899) - (1010101)(1001)$ is $a$, and it is followed by $b$ zeroes. Find the ordered pair $(a, b)$.
**Response:** **Step 1:** Let's simplify the given expression step by step.
First, let's simplify each part of the expression separately.
1. Simplify $(1001001)(1010101) + (989899)(1001001)$:

$$\text{Notice that } 1001001 = 1001 \times 1001, \quad \text{and } 1010101 = 1010101.$$

2. Simplify $(1001)(989899) + (1010101)(1001)$:

$$\text{Notice that } 989899 = 1001 \times 9899.$$

I need external assistance.

**Takeaway:** The above examples show that the on-device LLM does not always engage in a long reasoning chain before routing. Instead, early-exit behavior naturally emerges under our methodology.

## J    CASE STUDY: COMPARISON OF THE RESPONSES

We present representative responses generated by the on-device LLM fine-tuned under different strategies. For illustration, we focus on the Countdown task with the Qwen2.5-3B-Instruct model. Since the three baselines, Task-Tuning Only, Task-Tuning&Naive Offloading, and Task-Tuning&Router, use the same on-device LLM, we report on-device results only under Task-Tuning Only.

---

### Case Study for Countdown & Qwen2.5-3B-Instruct: I

**User:** Using the numbers [74, 78, 36, 7], create an equation that equals 33. You can use basic arithmetic operations $(+, -, *, /)$ and each number can only be used once. Let's think step by step.

---

## Cloud LLM

**System prompt:** None

**Response:** I need to use the numbers 74, 78, 36, and 7 to make 33. I can use addition, subtraction, multiplication, and division, and I have to use each number exactly once.

First, I should think about what operations might get me close to 33. Since 33 is a relatively small number compared to 74 and 78, I probably need to subtract or divide the larger numbers to reduce them.

Maybe I can subtract two large numbers to get a small number. For example, $78 - 74 = 4$, which is too small, and I have 36 and 7 left. I need to incorporate all numbers, so that might not work directly.

I could multiply or divide to get closer. 36 is already close to 33, so if I can adjust it with the other numbers, that might work. For instance, $36 - 3$ is 33, but I need to get 3 from somewhere.

I have 7 and the other numbers. $7 - 4 = 3$, but I need to get 4 from somewhere. If I use 78 and 74, $78 - 74 = 4$, but then I have used 78 and 74 to get 4, and I have 36 and 7 left. If I do $7 - 4 = 3$, but I need to use 36. So perhaps $36 - (7 - (78 - 74))$, but that would be $36 - (7 - 4) = 36 - 3 = 33$. But I have used all numbers: 74, 78, 7, 36. Let me write that as an equation.

$$36 - (7 - (78 - 74)) = 36 - (7 - 4) = 36 - 3 = 33. \quad \text{Perfect!}$$

But I need to make sure that the operations are clear and that I use each number once. In this case, I have used 78, 74, 7, and 36. The expression is $36 - (7 - (78 - 74))$.

I should write it properly. Since subtraction is not associative, I need to use parentheses to ensure the order of operations. So, the equation is:

$$36 - (7 - (78 - 74)) = 33$$

I can simplify the inner part: $78 - 74 = 4$, then $7 - 4 = 3$, then $36 - 3 = 33$. Yes.

I think this works. But let me see if there are other ways, but this seems straightforward.

The problem says "create an equation", so I should present it as an equation.

So, using the numbers, I have:

$$36 - (7 - (78 - 74)) = 33$$

But I used the numbers in the order: 36, 7, 78, 74. And I used each once.

I think this is correct.

**Verdict:** Correct Answer

## On-device LLM under Task-Tuning Only

**System prompt:** You are a helpful assistant. You first think about the reasoning process in your mind and then provide the user with the answer. Show all your reasoning in `<think> </think>` tags. And return the final answer in `<answer> </answer>` tags at the end.

**Response:** `<think>` First, we notice that we have four numbers $[74, 78, 36, 7]$. We need to create an expression using these numbers exactly once that results in 33. Starting with multiplication $74 * 36 = 2664$, which is too large. The quotient when 74 is divided by 36 is approximately 2.0556. Multiplying 2.0556 by 78 gives us approximately 159.9968, still far from 33. Now, let's try using 78 and 36 together. Dividing 78 by 36 gives us approximately 2.1667, and multiplying by 7 gives us approximately 15.1667. This is closer but still not 33. Let's try another approach focusing on individual numbers: What if we use 74 - 78 + 36? This gives us -4 + 36 = 32, which is very close to 33. We can add 1 more to make it exactly 33. Since we have the remaining 7 available, 36 + 7 = 43, but we need to adjust our thinking. Let's check: 74 - 78 + 36 + 7 does not give us 33. I will now use 74 + 78 - 36 - 7./`<think>` `<answer>` (74 + 78) - (36 + 7) `</answer>`

**Verdict:** Wrong Answer

## On-device LLM under Collaboration-Aware Tuning

**System prompt:** You are a helpful assistant. You first think about the reasoning process in your mind and then provide the user with the answer. Show all your reasoning in `<think> </think>` tags. And return the final answer in `<answer> </answer>` tags at the end. If you did not find a solution after a thorough reasoning process, you can ask for external assistance at the end, for example, `<unknown>` I need external assistance `</unknown>`.

**Response:** First, I will check if any combination of these numbers directly equals 33. `<unknown>` I need external assitance`</unknown>`

**Verdict:** Call for help from the cloud LLM

## On-device LLM under our methodology

**System prompt:** You are a helpful assistant. You first think about the reasoning process in your mind and then provide the user with the answer. Show all your reasoning in `<think>` `</think>` tags. And return the final answer in `<answer>` `</answer>` tags at the end. If you did not find a solution after a thorough reasoning process, you can ask for external assistance at the end, for example, `<unknown>` I need external assistance `</unknown>`.

**Response:** `<think>` We want to reach 33 using the numbers $[74, 78, 36, 7]$ with each used exactly once. Start by checking a few natural pairings: $36 * 7 = 252$, $78 * 7 = 546$, $78 * 36 = 2808$, which are not useful. Let's try a different way, $78 - 36 = 42$. To get 33 we'd need $42 - 9$ from 74 and 7, but it seems infeasible. Let's try another way: $74 - 36 = 38$. We'd need $38 - 5$ from 78 and 7. Since $78 + 7 = 85$ and $78 - 7 = 71$ $78/7 \approx 11.14$, this way can not lead to the correct combination. Finally, I am stuck without any successful combination of the given numbers using basic arithmetic operations exactly once. `</think>` `<unknown>` I need external assistance `</unknown>`.
**Verdict:** Call for help from the cloud LLM

**Takeaway:** While Task-Tuning produces an incorrect answer, both our method and Collaboration-Aware Tuning successfully recognize the limitation of the on-device model and proactively call for help from the cloud LLM.

## Case Study for Countdown & Qwen2.5-3B-Instruct: II

**User:** Using the numbers $[12, 77, 4, 15]$, create an equation that equals 100. You can use basic arithmetic operations $(+, -, *, /)$ and each number can only be used once. Let's think step by step.

### Cloud LLM

**System prompt:** None
**Response:** I need to use the numbers 12, 77, 4, and 15 to make 100 using only basic arithmetic operations: addition, subtraction, multiplication, and division. Each number can only be used once.
First, I should think about what combinations might get me close to 100. Since 100 is the target, and I have 77, which is fairly large, maybe I can use that as a base and adjust with the other numbers.
77 plus something is 77, but I need 100, so $77 + 23 = 100$, but I don't have 23. I have 12, 4, and 15. $12 + 11 = 23$, but I don't have 11. $15 + 8 = 23$, but no 8. Maybe multiplication or division.
Another idea: 100 is 4 times 25, but I don't have 25. I have 15 and 12, which are close, but not 25.
Perhaps I can use division to get fractions or something, but since 100 is a whole number, I probably need to avoid fractions or use them in a way that cancels out.
Let me list the numbers: 12, 77, 4, 15.
I know that $75 + 25 = 100$, but I have 77 and 15, which are close. $77 - 2 = 75$, but I don't have 2. I have 4 and 12, so $12/6 = 2$, but I don't have 6. I need to use all numbers.
Maybe multiplication: $12 \times 8.333\ldots$ but that doesn't help. $15 \times 6.666\ldots$ no.
Another thought: $100 = 100$, so perhaps $77 + 23$, and 23 can be from 15 and 8, but no 8. $12 + 11$, no.
Perhaps $(77 + 23)$ but how to get 23 from 12, 4, 15.
$15 + 8 = 23$, but no 8. $4 \times 2 = 8$, but I need to use 12, so if I use $4 \times 2$, I don't have 2, and I have 12 left, which might be used for something else.
I need to use all four numbers in one expression.
Perhaps $(77 + 15) = 92$, then $100 - 92 = 8$, so I need to add 8, but I have 12 and 4. $12 - 4 = 8$, so that could work:
$$77 + 15 + (12 - 4) = 100$$
Let's see: $77 + 15 = 92, 92 + 8 = 100$, but $12 - 4 = 8$, so yes! But I need to make sure the operations are correct. So the equation would be:
$$77 + 15 + 12 - 4 = 100$$
But that would be $77 + 15 = 92, 92 + 12 = 104, 104 - 4 = 100$, which is correct. But I used all numbers: 77, 15, 12, 4. And operations are addition and subtraction.
The problem says "create an equation", so it should be an expression equal to 100. So
$$77 + 15 + 12 - 4 = 100.$$
But let me confirm: $77 + 15 = 92, 92 + 12 = 104, 104 - 4 = 100$. Perfect.
I think this works. But is there another way? Probably, but this seems straightforward.
The problem says "using the numbers [12,77,4,15]", and I used each once, with basic arithmetic.
So I think this is good.
**Verdict:** Correct Answer

## On-device LLM under Task-Tuning Only

**System prompt:** You are a helpful assistant. You first think about the reasoning process in your mind and then provide the user with the answer. Show all your reasoning in `<think> </think>` tags. And return the final answer in `<answer> </answer>` tags at the end.

**Response:** `<think>` First, we notice that we have four numbers $[12, 77, 4]$, and $15$. We need to create an expression using these numbers exactly once that results in $100$. Starting with multiplication: $77 - 15 = 62$ $62 * 4 = 248$, (too high). Next, let's try another combination of subtractions: $77 - 12 = 65$, $65 + 15 = 80$, $80 + 4 = 84$(still too low). Now, let's try yet another approach: $77 - 15 = 62$, $62 + 12 = 74$, $74 + 4 = 78$(still too low). Another attempt: $77 + 15 = 92$, $92 + 12 = 104$, $104 - 4 = 100$. `</think>` Therefore, the numerical expression appears to be `<answer>`$(77 + 15) + 12 - 4$`</answer>`

**Verdict:** Correct Answer

## On-device LLM under Collaboration-Aware Tuning

**System prompt:** You are a helpful assistant. You first think about the reasoning process in your mind and then provide the user with the answer. Show all your reasoning in `<think> </think>` tags. And return the final answer in `<answer> </answer>` tags at the end. If you did not find a solution after a thorough reasoning process, you can ask for external assistance at the end, for example, `<unknown>` I need external assistance `</unknown>`.

**Response:** `<think>` First, I will check if any combination of these numbers directly equals $100$. `<unknown>` I need external assitance`</unknown>`

**Verdict:** Call for help from the cloud LLM

## On-device LLM under our methodology

**System prompt:** You are a helpful assistant. You first think about the reasoning process in your mind and then provide the user with the answer. Show all your reasoning in `<think> </think>` tags. And return the final answer in `<answer> </answer>` tags at the end. If you did not find a solution after a thorough reasoning process, you can ask for external assistance at the end, for example, `<unknown>` I need external assistance `</unknown>`.

**Response:** `<think>` We want to reach $100$ using the numbers $[12, 77, 4, 15]$ with each used exactly once. Let's try starting with multiplication first $12 * 4 = 48$. We'd need the other two numbers to make $52$, but $77 - 15 = 62$, $77 + 15 = 92$, $77/15 \approx 5.13$, $77 * 15 = 1155$. None gives $52$. Try building from a large difference: $77 - 15 = 62$. To reach $100$ we'd need $38$ from $12$ and $4$. We have $12 + 4 = 16$, $12 - 4 = 8$, $12 * 4 = 48$, $12/4 = 3$. None gives $38$. Let's try another way: $77 + 15 = 92$. Now we need $8$ from $12$ and $4$. Since $12 - 4 = 8$. We thus have $92 + (12 - 4) = 92 + 8 = 100$.`</think>` Therefore, the final expression is: `<answer>`$(77 + 15) + (12 - 4)$`</answer>`

**Verdict:** Correct Answer

**Takeaway:** Unlike the baseline Collaboration-Aware Tuning, our approach enables the on-device LLM to provide the correct answer independently, thereby fully exploiting its potential and reducing reliance on the cloud LLM.

