# OpenReview forum: "Collaborative Device-Cloud LLM Inference through Reinforcement Learning"
_ICLR.cc/2026/Conference — Submitted to ICLR 2026_

### Official Review · Reviewer_yewN · 2025-10-29

**Soundness:** 3
**Presentation:** 3
**Contribution:** 2
**Rating:** 2
**Confidence:** 4

**Summary:**

This paper presents a novel framework for collaborative device-cloud LLM inference that eliminates the need for an external router. The core idea is to empower the on-device LLM to autonomously decide when to offload a query. This is achieved through a unified reinforcement learning (RL) framework where the model is post-trained to either solve a task or "call for help" after attempting a solution. A hierarchical reward function guides this learning process, and a new Group-Adaptive Policy Gradient (GAPG) algorithm is proposed for stable optimization under a cloud usage budget. Experiments demonstrate that this integrated approach significantly outperforms baseline methods and substantially narrows the performance gap to a full cloud-LLM solution.

**Strengths:**

- The paper introduces a highly novel and elegant paradigm by embedding the routing logic directly into the on-device LLM, which eliminates the need for a separate, external router and simplifies the overall system architecture.
- The proposed Group-Adaptive Policy Gradient (GAPG) algorithm is well-motivated, addressing a specific failure mode of existing RL methods in this hierarchical reward setting , and the adaptive prompt filtering is a practical mechanism for handling budget constraints.
- The empirical evaluation is thorough, demonstrating consistent outperformance over strong baselines across multiple models and benchmarks. The results effectively validate the practical benefits of the proposed unified training framework.

**Weaknesses:**

- The claimed novelty of the core gradient estimator in the GAPG algorithm is questionable. The key modification, removing standard deviation normalization from the GRPO advantage calculation, appears to be identical to the one proposed in the concurrent work "Dr. GRPO" (ICML Workshop, https://arxiv.org/abs/2503.20783), which is not cited or discussed in this paper.
-The experiments are confined to mathematical reasoning tasks. The framework's applicability to more complex tasks (e.g., code generation, etc.) is not demonstrated, and designing an effective reward signal for such tasks could be a significant challenge.
- The routing decision is made only after the on-device model has completed its full generation process. This "post-hoc" decision-making process means that for any query ultimately offloaded to the cloud, the initial on-device computation is entirely wasted, leading to inefficient use of device resources and increased end-to-end latency for the user.
- The collaboration mechanism is a simple binary hand-off. The on-device model cannot pass its partial reasoning to the cloud model, which misses an opportunity for a more nuanced and potentially more efficient collaboration.

**Questions:**

- The core modification in your GAPG algorithm (using r - mean(r) for advantage) appears to be identical to the one proposed in the concurrent work "Dr. GRPO." Could you please clarify the relationship between your work and this prior art and better position your contribution?
- The routing decision is made at the end of the generation process. Does this mean the on-device model always completes its full reasoning, even if it determines early on that a solution is unlikely? Have you explored mechanisms for an "early exit" to save on-device computation?
- How sensitive is the model's final performance and routing behavior to the specific weights chosen for the hierarchical reward function (i.e., $\alpha_a, \alpha_c, \alpha_f$)? A sensitivity analysis would help understand the robustness of the reward design.

---

> ### Author Response · Authors · 2025-11-23
> **Responses (1/3)**
>
> We thank the reviewer for the time, effort, and valuable feedback.
>
> ### **W1-1. Novelty of the core gradient estimator in the GAPG algorithm: the key modification, removing standard deviation normalization from the GRPO advantage calculation, appears to be identical to the one proposed in the concurrent work "Dr. GRPO"** & **Q1. Could you please clarify the relationship between your work and this prior art and better position your contribution?**
>
> We thank the reviewer for the reminder. We were unaware of the concurrent Dr. GRPO work the reviewer mentioned during the preparation of the manuscript. We acknowledge that the advantage computation we use matches the one adopted in Dr. GRPO. This overlap arises naturally because **removing the standard-deviation normalization is a common practice in policy-gradient methods**, including the basic REINFORCE-style policy gradient. However, the overall **CAPG algorithm is different** from *Dr. GRPO* in several key aspects.
>
> First, the **key components** of CAPG include the **hierarchical reward design**, **adaptive prompt filtering**, and the **removal of standard-deviation normalization**. Our motivation for identifying the issue of standard-deviation normalization in the GRPO primarily stems from the **hierarchical reward formulation**, which, together with adaptive prompt filtering, is unique to our method.
>
> Second, Dr. GRPO follows the GRPO pipeline and modifies its advantage calculation, whereas **CAPG is more of a online policy gradient** method. Consequently, CAPG does not rely on clipping or importance sampling, both of which are inherited by GRPO-style algorithm. The removal of standard-deviation normalization arises naturally from the **policy-gradient derivation**.
>
> Third, and more importantly, the **core contribution of our work is not a new stand-alone RL algorithm**. The primary novelty lies in formulating and solving the **on-device–cloud LLM collaboration problem** in an end-to-end fashion, which, based on our understanding, has *not* been explored in prior literature. Our contributions are three-fold:
>
> 1. **Collaboration paradigm.** We propose the first principled approach to integrate routing behavior *directly* into the on-device LLM through RL-based post-training, enabling the model to learn when to rely on its own reasoning and when to offload to the cloud LLM.
>
> 2. **Problem formulation.** We introduce a hierarchical reward design that jointly captures task correctness and device-cloud coordination. This formulation is unique to the collaborative inference setting and is essential for this methodology.
>
> 3. **Algorithmic design.** We develop a policy-gradient method tailored to this formulation, featuring **group-response sampling** (inspired by GRPO but not equivalent to it and its variants, Dr. GRPO) and **adaptive prompt filtering**, both of them are specific to our collaboration inference problem.
>
> Overall, this work is **application-oriented**: its primary objective is develop the first end-to-end training framework for collaborative device-cloud reasoning.
>
> > We have included **Appendix B** in the revised manuscript to discuss Dr. GRPO and further clarify our contributions.
>
> ### **W1-2. The framework's applicability to more complex tasks is not demonstrated, and designing an effective reward signal for such tasks could be a significant challenge.**
>
> We agree with the reviewer that designing reward functions for more complex tasks is inherently challenging. However, this challenge is **not specific to our method**, it is a well-recognized problem across the entire reinforcement-learning-with-verified-rewards (RLVR) family of approaches. Existing RLVR-based post-training studies are almost exclusively evaluated on tasks with objective correctness signals, such as mathematics reasoning.
>
> In this context, our work follows the same established practice. We intentionally focus on mathematical and arithmetic reasoning tasks because they provide **deterministic, verifiable reward signals**. Our contribution is to make the **first attempt** at an end-to-end optimization framework for on-device–cloud collaborative inference, with a particular focus on enabling **collaborative reasoning**, which is both a popular  application and a challenging task for on-device LLMs. Extending collaborative device–cloud RL training to more complex tasks would require **reward-model-based evaluation** or **LLM-as-a-Judge mechanisms**, which is an important direction for future work.
>
> > The limitation noted by the reviewer is **a limitation of the RLVR regime**, not of our framework specifically. Our method inherits the same scope as other RLVR methods and can be applied to the tasks where deterministic reward signals exist. To clarify this, we revised the manuscript and added a discussion on the potential limitation of our work.

---

> ### Author Response · Authors · 2025-11-23
> **Responses (2/3)**
>
> ### **W2. The routing decision is made only after the on-device model has completed its full generation process. This "post-hoc" decision-making process means that for any query ultimately offloaded to the cloud, the initial on-device computation is entirely wasted, leading to inefficient use of device resources and increased end-to-end latency for the user.**
>
> It is **difficult** to make reliable routing decisions from the query alone: as we showed, a standalone binary classifier performs poorly, and the on-device model’s reasoning provides far stronger signals than query-only features. The **goal of this work** is to improve the inference accuracy of device-cloud LLM collaboration system, and our method consistently achieves higher accuracy than query-only routing. The device computation is therefore intentionally used to **trade latency and resource for accuracy**, a design choice that aligns with a broad set of recent LLM techniques, such as *long reasoning chains, test-time scaling, LLM cascades, and multi-LLM debate*, where additional computation is leveraged to improve final performance.
>
> In addition, we showed in our response to **Question 2**, on-device LLM does not always complete a long reasoning trace before routing.
>
> ### **W3. The on-device model cannot pass its partial reasoning to the cloud model, which misses an opportunity for a more nuanced and potentially more efficient collaboration.**
>
> The cloud LLM is typically **far more capable** than the on-device model, and its reasoning style differs substantially. As a result, forwarding the device’s partial reasoning to the cloud rarely improves performance. Moreover, transmitting partial reasoning adds unnecessary communication overhead and shifts additional burden onto the device.
>
> Additionally, our collaboration strategy can be naturally extended to the scenario where the device sends **both the query and its response** to the cloud. We implemented this variant and evaluated its performance, and the results are shown below.
>
> **Qwen2.5-1.5B-Instruct**
>
> | Transmitting           | MATH-lighteval | MATH-500 | AMC23 | AIME24 | MinervaMath | Avg. |
> | ---------------------- | -------------- | -------- | ----- | ------ | ----------- | ---- |
> | Query only             | 78.0           | 81.6     | 57.5  | 23.3   | 40.8        | 56.2 |
> | Query & Local response | 78.1           | 81.4     | 57.5  | 23.3   | 41.0        | 56.3 |
>
> **Llama-3.2-3B-Instruct**
>
> | Transmitting           | MATH-lighteval | MATH-500 | AMC23 | AIME24 | MinervaMath | Avg. |
> | ---------------------- | -------------- | -------- | ----- | ------ | ----------- | ---- |
> | Query only             | 79.5           | 68.6     | 52.5  | 33.3   | 43.4        | 55.5 |
> | Query & Local response | 79.7           | 68.5     | 52.5  | 33.3   | 43.4        | 55.5 |
>
> As can be observed from the above tables, sending the response from the on-device LLM doesn't siginificantly benefit the overall accuracy. The main reason is that the cloud LLM is is far more powerful than the on-device LLM. Importantly, our framework can **support scenarios where passing partial reasoning is desirable.**
>
> ### **Q1. The core modification in your GAPG algorithm (using r - mean(r) for advantage) appears to be identical to the one proposed in the concurrent work "Dr. GRPO." Could you please clarify the relationship between your work and this prior art and better position your contribution?**
>
> We thank the reviewer for raising this point. We discussed the relationship between our work and Dr. GRPO in our response to **W1-1**.

---

> ### Author Response · Authors · 2025-11-23
> **Responses (3/3)**
>
> ### **Q2. The routing decision is made at the end of the generation process. Does this mean the on-device model always completes its full reasoning, even if it determines early on that a solution is unlikely? Have you explored mechanisms for an "early exit" to save on-device computation?**
>
> We thank the reviewer for this insightful question. The routing decision is made autonomously by the on-device LLM, and **it does *not* always complete a full reasoning trace**. When the model detects that a correct solution is unlikely, it **terminates its reasoning early and calls for help**. Hence, **early exit** already emerges under our current methodology, without requiring additional design.
>
> To illustrate this phenomenon, we present two representative **call-for-help (early-exit) patterns** the on-device model exhibits in practice.
>
> (1). For what real values of $x$ is $-4<x^{4}+4x^{2}<21$ satisfied? Express your answer in interval notation.
>
> **Raw output:** Step 1: We need to solve the inequality \\(-4 < x^4 + 4x^2 < 21\\).\n\nStep 2: Let's start by solving the left part of the inequality, \\( -4 < x^4 + 4x^2 \\).\n\nStep 3: Let's solve the right part of the inequality, \\( x^4 + 4x^2 < 21 \\).\n\nI need external assistance.
>
> (2). The rightmost non-zero digit in  $(1001001)(1010101)+(989899)(1001001)-(1001)(989899)-(1010101)(1001)$ is $a$, and it is followed by $b$ zeroes. Find the ordered pair $(a,b)$.
>
> **Raw output:** Step 1: Let's simplify the given expression step by step.\n\nFirst, let's simplify each part of the expression separately.\n\n1. Simplify \\((1001001)(1010101) + (989899)(1001001)\\):\n   - Notice that \\(1001001 = 1001 \\times 1001\\), and \\(1010101 = 1010101\\).\n\n2. Simplify \\((1001)(989899) + (1010101)(1001)\\):\n   - Notice that \\(989899 = 1001 \\times 9899\\).\n\nI need external assistance.
>
> In addition, we provide a **concrete comparison of the token cost** of on-device LLM between calling for help and solving the problem independently.
>
> | Token cost | Calling for help | Correct answers (Locally) |
> | - | - | - |
> | Llama-3.2-1B-Instruct | 83| 234  |
> | Qwen2.5-1.5B-Instruct | 269 | 325|
> | Llama-3.2-3B-Instruct | 141 | 343|
>
> As shown in the table, the on-device model typically requires **fewer tokens to detect that a problem is beyond its capability than to solve it fully**.
>
> > In the revised manuscript, we have included **Appendix I**, where we show several concrete call-for-help patterns under our methodology. These examples demonstrate that the on-device model **does not always complete its full reasoning** before calling assistance.
>
> ### **Q3. How sensitive is the model's final performance and routing behavior to the specific weights chosen for the hierarchical reward function?**
>
> The model's performance and routing behavior are sensitive to the reward weights. In particular, to realize a desired call-for-help ratio, the **primary tunable hyperparameters** are the weights $\alpha_a$, $\alpha_f$ and $\alpha_c$. As the format reward is smaller than coordination reward $\alpha_c$, the **dominant factor** governing the trade-off between independent solving and cloud assistance is the **relative magnitude** of $\alpha_a $ versus $\alpha_c$. Different combination of $\alpha_a$ and $\alpha_c$ will result in different call-for-help ratios. Moreover, for different models, the combination will be different to attain the same call-for-help ratio. As we reported in Table 3 of the original manuscript, the coordination reward for different model are set differently.
>
> To show this phenomenon, we show the call for help ratio under different combination of $\alpha_a$ and $\alpha_c$. The results are obtained based on Qwen2.5-1.5B-Instruct model and MATH-lighteval task. The format reward $\alpha_f$ is set 0. Note that the accuracy shown below doesn't consider the call-for-help budget constraint.
>
> | Reward weights | Call-for-help ratio | Local accuracy | Overal accuracy (without budget constraint) |
> | -| - | - | - |
> | $\alpha_a = 1,\alpha_c=0$    | 0  | 56.1 | 56.1 |
> | $\alpha_a = 1,\alpha_c=0.3$  | 29.7  | 48.9 | 78.0 |
> | $\alpha_a = 1,\alpha_c=0.35$ | 42.1 | 47.4 | 88.7|
> | $\alpha_a = 1,\alpha_c=0.45$  | 50.3 | 43.1  | 92.3|
>
> We can see from the table, increase $\frac{\alpha_c}{\alpha_a}$ makes the model focus more on call for help and pay less attention to improve its independent problem solving ability. To balance call for cloud assistance, the tuning of the reward is critical, which further illustrate the importance of hierarchical reward.
>
> > Different choices of $\alpha_a$ and $\alpha_c$ yield different different routing behavior, offering **flexibility** to **adapt to different target calling-for-help ratios**. We added this ablation study in **Appendix C** of the revised manuscript.
>
> Again, we appreciate the reviewer’s time and thoughtful comments.

---

### Official Review · Reviewer_Xpht · 2025-10-30

**Soundness:** 2
**Presentation:** 3
**Contribution:** 2
**Rating:** 4
**Confidence:** 4

**Summary:**

This paper proposes a reinforcement learning framework for collaborative device-cloud LLM inference where lightweight on-device models learn to autonomously decide when to offload queries to powerful cloud LLMs. Unlike existing approaches using external routers, the method integrates routing capability directly into post-training through a reward maximization problem with hierarchical rewards. The authors develop a Group-Adaptive Policy Gradient (GAPG) algorithm featuring unbiased group-level gradient estimation and adaptive prompt filtering to balance local problem-solving with cloud assistance. Experiments across mathematical benchmarks demonstrate the approach consistently outperforms baselines, maintains stable training, and significantly narrows the performance gap to full cloud LLM usage while respecting offloading constraints.

**Strengths:**

+ This work elegantly integrates routing optimization directly into the post-training process, eliminating the need for separate external routers. This unified approach allows the on-device LLM to simultaneously improve its problem-solving capabilities while learning judicious offloading strategies.
+ The group-level policy gradient addresses GRPO's limitations in collaborative settings, while adaptive prompt filtering effectively enforces cloud usage constraints, preventing the model from over-relying on external assistance during training.
+ The evaluation demonstrates performance across multiple model sizes (1B to 3B parameters) and diverse mathematical benchmarks.

**Weaknesses:**

- The work lacks clear motivation and validation for why routing capability should be embedded in the on-device LLM with a first-device-then-cloud cascade way rather than using separate routers for directly choosing which LLM. The optimization goal appears vague—it achieves significantly lower accuracy than cloud LLM alone (cf. Table 1), yet doesn't explicitly quantify or optimize the cost trade-off between on-device computation and cloud invocation, making it difficult to assess whether the approach provides genuine practical benefits. In particular, accuracy should be the first objective in real-world deployment.

- The core technical contribution is incremental. The group-level policy gradient (Proposition 3.1) closely resembles existing GRPO with minor modifications. The only novel component—adaptive prompt filtering—is a simple heuristic that selects training prompts based on response correctness ratios. This lacks theoretical justification and appears to be an ad-hoc empirical design without principled foundations for why this specific filtering strategy is optimal.

- Evaluation is restricted exclusively to mathematical reasoning tasks (Countdown and MATH benchmarks). The paper completely ignores other important domains like general question answering, commonsense reasoning, coding, summarization, or multi-turn dialogue. This severely limits claims about generalizability, as mathematical problems may have unique characteristics that don't transfer to other collaborative inference scenarios where routing decisions differ fundamentally.

- The paper compares against only one routing method (Task-Tuning&Router using DeBERTa) and naive random offloading, missing numerous recent works in LLM routing and cascade systems. Notable omissions include FrugalGPT, RouteLLM variants, context-aware cascading policies, and ensemble-based routing methods cited in related work but never experimentally compared, weakening claims of superiority.

- The paper provides no ablation analysis to isolate the contribution of individual components. Critical questions remain unanswered: How much does hierarchical reward design contribute versus the gradient estimator? What's the impact of different reward weights (αₐ, αc, αf)? How sensitive is performance to the filtering ratio ρ? Without ablations, it's impossible to understand which design choices actually matter or whether simpler alternatives might work equally well.

**Questions:**

Please see weaknesses.

---

> ### Author Response · Authors · 2025-11-23
> **Responses (1/4)**
>
> We thank the reviewer for the time, effort, and valuable feedback.
>
> ### **W-1. Motivation: why routing capability should be embedded in the on-device LLM with a first-device-then-cloud cascade way rather than using separate routers**
>
> We discussed in the limitations of the router method in **Section 2.2** of the original manuscript. First, the router is essentially a binary classifier. It is inherently difficult for such a classifier to judge whether the on-device LLM can solve a problem based solely on the prompt's surface pattern, since problems with similar structures may vary greatly in difficulty.  Additionally, as we can see in the experiment section, binary classifier based router performs poorly. Conversely, using a more powerful LLM with reasoning ability as the router would be inefficient and wasteful, since making a routing decision would require duplicating the reasoning process that should be performed by the on-device LLM. This redundancy adds unnecessary computation and storage overhead.
>
> ### **W1-2. It achieves significantly lower accuracy than cloud LLM alone (Table 1), yet doesn't explicitly quantify or optimize the cost trade-off between on-device computation and cloud invocation**
>
> We would like to emphasize that the performance reported in Table 1 is evaluated under a fixed call-for-cloud budget of $30 \\%$. The goal of this paper is not to outperform the cloud LLM, but to demonstrate that **given the same cloud-call budget**, our proposed method achieves the strongest accuracy among all baselines.
>
> Furthermore, Figure 5 shows how accuracy evolves as the call-for-cloud ratio increases. As the budget increases, all methods improve, but our proposed method approximately math cloud LLM's performance at only $60 \\%$ cloud calls, which is the lowest ratio required among all baselines. This highlights that our approach provides **the most favorable accuracy–cost trade-off** under practical deployment constraints. Moreover, the call-for-help ratios can be tuned under our framework, which thus allows us to flexibly balance on-device computation against cloud invocation.
>
> ### **W2. The core technical contribution is incremental. The group-level policy gradient closely resembles existing GRPO with minor modifications. The only novel component, adaptive prompt filtering, is a simple heuristic that selects training prompts based on response correctness ratios. This lacks theoretical justification and appears to be an ad-hoc empirical design without principled foundations for why this specific filtering strategy is optimal.**
>
> We would like to clarify that the core contribution of this work is **not** proposing a new stand-alone RL algorithm. Instead, the technical novelty lies in formulating and solving the **on-device–cloud LLM collaboration** problem in an end-to-end manner.  Our contributions are three-fold:
>
> 1. **Collaboration paradigm.** We propose the first principled approach to integrate routing behavior *directly* into the on-device LLM through RL-based post-training, enabling the model to learn when to rely on its own reasoning and when to offload to the cloud LLM.
>
> 2. **Problem formulation.** We introduce a hierarchical reward design that jointly captures task correctness and device-cloud coordination. This formulation is unique to the collaborative inference setting and is essential for this methodology.
>
> 3. **Algorithmic design.** We develop a policy-gradient method tailored to this formulation, featuring **group-response sampling** (inspired by GRPO) and **adaptive prompt filtering**, both of which are specific to the collaboration problem.
>
> > Overall, the contribution is **not** an incremental algorithmic improvement over GRPO, but rather the **first end-to-end training framework** for collaborative device–cloud inference.
>
> **Regarding adaptive prompt filtering**: We agree that the mechanism is simple, and we do not claim it is theoretically optimal. However, it is an **effective and practical design** for stabilizing training in our collaborative inference framework. This work is application-oriented, we prioritize the method that improve training stability and final performance. The empirical effectiveness of adaptive filtering is what motivates its use.

---

> ### Author Response · Authors · 2025-11-23
> **Responses (2/4)**
>
> ### **W3. Evaluation is restricted exclusively to mathematical reasoning tasks. The paper completely ignores other important domains like general question answering, commonsense reasoning, coding, summarization, or multi-turn dialogue. This severely limits claims about generalizability, as mathematical problems may have unique characteristics that don't transfer to other collaborative inference scenarios where routing decisions differ fundamentally.**
>
> We agree with the reviewer that designing reward functions for open-ended or subjective tasks is inherently challenging. However, this challenge is **not specific to our method**, it is a well-recognized limitation across the entire reinforcement-learning-with-verified-rewards (RLVR) family of approaches. Existing RLVR-based post-training studies (e.g., GRPO and GRPO variants) are almost exclusively evaluated on tasks with objective correctness signals, such as mathematics reasoning. Reliable reward design for open-ended tasks remains an open research problem.
>
> In this context, our work follows the same established practice. We intentionally focus on mathematical and arithmetic reasoning tasks because they provide **deterministic, verifiable reward signals** essential for evaluating the effectiveness of a new collaborative training paradigm. Our contribution is to make the **first attempt** at an end-to-end optimization framework for on-device–cloud collaborative inference, with a particular focus on enabling **collaborative reasoning**.
>
> We would like to clarify that **we do not claim that our methodology currently applies to all categories of tasks**. Extending collaborative device–cloud RL training to more complex or subjective tasks would require **reward-model-based evaluation** or **LLM-as-a-Judge mechanisms**, which is an important direction for future work.
>
> > The limitation noted by the reviewer is **a limitation of the RLVR regime**, not of our framework specifically. Our method inherits the same scope as other RLVR methods and can be applied to the tasks where deterministic reward signals exist. To clarify this, we revised the manuscript and added a discussion on the potential limitation of our work.
>
> ### **W4. Only one routing method, notable omissions include FrugalGPT, RouteLLM variants, context-aware cascading policies, and ensemble-based routing methods**
>
> We would like to clarify that some works mentioned by the reviewer are **not applicable** to the device–cloud LLM collaboration setting studied in this paper, and some of them are actually **similar** as the baseline we considered in our original manuscript.
>
> **FrugalGPT [R1]** also adopt a router to select LLMs. However, it select multiple LLMs for one query and sequentially ask these LLMs to generate response until a target quality is reached. This is fundamentally different from our device–cloud collaboration setup.
>
> **Context-aware cascading policy [R2]** extend FrugalGPT by adding contextual features but still operate in the same **multi-LLM cascade** regime, where several LLMs may be queried sequentially for the same prompt, making it not an appropriate baseline.
>
> **Ensemble-based routing method [R3]** also assumes access to several diverse LLMs and predicts a routing probability distribution over them. This amounts to a **multiclass classification problem across**, which is not aligned with our binary device–cloud setting. Conceptually, this is similar to the baseline we already included (Task-Tuning&Router).
>
> **RouteLLM [R4]** take BERT-base and LLaMA3-8B as the classifier to route the query between a small model and powerful model, which is essentially similar to the baseline we already included (Task-Tuning&Router).
>
> [R1] "Frugalgpt: How to use large language models while reducing cost and improving performance." *2023*.
>
> [R2] "Efficient contextual llm cascades through budget-constrained policy learning." *2024*.
>
> [R3] "Routing to the expert: Efficient reward-guided ensemble of large language models." *2024*.
>
> [R4] "Routellm: Learning to route llms with preference data." *2024*.

---

> ### Author Response · Authors · 2025-11-23
> **Responses (3/4)**
>
> ### **W5-1. Lack ablation analysis. How much does hierarchical reward design contribute versus the gradient estimator?**
>
> **Hierarchical reward**
>
> Hierarchical reward is **central to our problem formulation**. If we assign the *same* reward to both “calling for help” and “producing a correct answer,” the model quickly collapses to always calling for help, because this pattern is trivial to learn and consistently yields high reward. To demonstrate this, we report the call-for-help ratio of our algorithm **with and without** hierarchical reward.
>
> **Qwen2.5-1.5B-Instruct and MATH-lighteval task (Call-for-help ratio, %)**
>
> | Training steps              | 0    | 5    | 10   | 15   | 20   | 25   | 30   | 35   | 40   |
> | --------------------------- | ---- | ---- | ---- | ---- | ---- | ---- | ---- | ---- | ---- |
> | Without hierarchical reward | 6.2  | 28.7 | 56.5 | 75.3 | 85.9 | 86.4 | 87.8 | 86.2 | 87.6 |
> | With hierarchical reward    | 6.2  | 17.9 | 23.1 | 22.1 | 25.4 | 26.2 | 22.4 | 23.1 | 24.7 |
>
> **Qwen2.5-3B-Instruct and Countdown task (Call-for-help ratio, %)**
>
> | Training steps              | 0    | 5    | 10   | 15   | 20   | 25   | 30   | 35   | 40   |
> | --------------------------- | ---- | ---- | ---- | ---- | ---- | ---- | ---- | ---- | ---- |
> | Without hierarchical reward | 10.7 | 89.2 | 100  | 100  | 100  | 100  | 100  | 100  | 100  |
> | With hierarchical reward    | 10.7 | 21.8 | 22.5 | 19.7 | 28.2 | 32.7 | 36.8 | 32.1 | 30.9 |
>
> As shown in the tables above, removing the hierarchical reward causes the on-device model to rapidly **exploit this bug** and overwhelmingly call for help, *hacking the reward*. These results highlight that hierarchical reward is a **core component** required for the collaborative routing mechanism to work.
>
> **Gradient estimator**
>
> We formulate a **reward maximization** problem and apply a **basic policy gradient method** enhanced with **group response sampling**, inspired by GRPO. To validate the effectiveness of  the **group-based gradient estimator** , we vary the **group size** and examine how the performance changes.  In the extreme case where ($G = 1$), the method collapses to standard policy gradient. The corresponding results are reported below.
>
> **Qwen2.5-1.5B-Instruct  (Accuracy %)**
>
> | Group size | MATH-lighteval | MATH-500 | AMC23 | AIME24 | MinervaMath | Avg. |
> | ---------- | -------------- | -------- | ----- | ------ | ----------- | ---- |
> | 4          | 56.0           | 61.3     | 35.0  | 23.3   | 22.1        | 39.5 |
> | 8          | 78.0           | 81.6     | 57.5  | 23.3   | 40.8        | 56.2 |
> | 16         | 78.2           | 81.1     | 57.5  | 23.3   | 41.2        | 56.3 |
>
> **Llama-3.2-3B-Instruct  (Accuracy %)**
>
> | Group size | MATH-lighteval | MATH-500 | AMC23 | AIME24 | MinervaMath | Avg. |
> | ---------- | -------------- | -------- | ----- | ------ | ----------- | ---- |
> | 4          | 53.2           | 53.9     | 27.5  | 23.3   | 33.5        | 38.3 |
> | 8          | 79.5           | 68.6     | 52.5  | 33.3   | 43.4        | 55.5 |
> | 16         | 79.4           | 69.2     | 52.5  | 33.3   | 44.1        | 55.7 |
>
> As we can see from this Table, reducing the group size to $G=4$ leads to a clear performance drop, which means that the **group sampling matters** in terms of the overal performance. Therefore, the **group-level gradient estimator is an essential component** of our framework.
>
> > In summary, both components, the **hierarchical reward** introduced in the problem formulation and the **group-based gradient estimator** developed for the algorithm, are essential for the proposed methodology. **This framework does not function properly when either component is missing.** We added an **Appendix C** in the revised manuscript to study the impact of hiearachical rewards weights and the gradient  estimator.

---

> ### Author Response · Authors · 2025-11-23
> **Responses (4/4)**
>
> ### **W5-2. What's the impact of different reward weights? How sensitive is performance to the filtering ratio?**
>
> ***Filtering ratio***
>
> The filtering ratio $\rho$ is **not** a hyperparameter, which adapts to a **system parameter** determined by Equation (2), the ratio of using on-device LLM to cloud LLM. By letting $\rho > 0$, the on-device model naturally receives both types of signals, **when to call for help** and **when to solve a problem independently**.  The role of filtering is to ensure that the on-device LLM strength *both* behaviors during training. While filtering determines whether both signals are present, the **relative strength** of these signals is primarily governed by the **reward weights**. This design choice simplifies the system: filtering provides the behavioral modes, and the reward weights control how strongly each mode is encouraged. Importantly, this does **not** diminish the role of adaptive filtering, its purpose is to ensure exposure to both behaviors and to provide coarse-level control, which is a **prerequisite** for adjusting their balance through reward weighting in a fine-grained manner.
>
> **Reward weights**
>
> To achieve a specific call-for-help ratio, the **main tunable hyperparameters** are the reward weights $\alpha_a$, $\alpha_f$ and $\alpha_c$. As the formatting reward $\alpha_f $ is smaller than the coordination reward $\alpha_c $, the **dominant factor** governing the trade-off between independent solving and cloud assistance is the **relative magnitude** of $\alpha_a $ versus $\alpha_c$. Moreover, different on-device models may require different choices of $ \alpha_a$ and $\alpha_c$ to achieve the **same** target call-for-help ratio. As reported in Table 3 of the main paper, the coordination rewards are adjusted per model to align them to similar call-for-help behavior.
>
> To further illustrate this, we report the call-for-help ratios obtained under different combinations of $\alpha_a$ and $\alpha_c$. The following results are based on the Qwen2.5-1.5B-Instruct model evaluated on MATH-lighteval. The formatting reward is fixed at $\alpha_f = 0$. Note that the accuracies shown below do not account for the call-for-help budget constraint.
>
> | Reward weights               | Call-for-help ratio | Local accuracy | Overal accuracy (without budget constraint) |
> | ---------------------------- | ------------------- | -------------- | - |
> | $\alpha_a = 1,\alpha_c=0$    | 0 | 56.1           | 56.1 |
> | $\alpha_a = 1,\alpha_c=0.3$  | 29.7| 48.9           | 78.0 |
> | $\alpha_a = 1,\alpha_c=0.35$ | 42.1 | 47.4           | 88.7|
> | $\alpha_a = 1,\alpha_c=0.45$  | 50.3 | 43.1  | 92.3|
>
> > Different combinations of $\alpha_a $ and $\alpha_c$ lead to different routing behavior, offering **flexibility** to **adapt to different target calling-for-help ratios**. We added an **Appendix C** in the revised manuscript to study the impact of hiearachical rewards weights.
>
> Again, we appreciate the reviewer’s time and thoughtful comments.

---

### Official Review · Reviewer_faeX · 2025-10-30

**Soundness:** 2
**Presentation:** 3
**Contribution:** 2
**Rating:** 4
**Confidence:** 3

**Summary:**

This paper focuses the efficiency-accuracy trade-off in device-cloud LLM collaboration by proposing a reinforcement learning-based unified framework that eliminates external routers.
The core idea is to train the on-device LLM to simultaneously enhance its problem-solving ability and learn routing strategies during post-training. Key innovations include: (1) a collaboration-aware hierarchical reward that guides the on-device LLM to call the cloud LLM (e.g., DeepSeek-R1) only when it cannot solve tasks independently; (2) a Group-Adaptive Policy Gradient (GAPG) algorithm with an unbiased group-level gradient estimator and adaptive prompt filtering (controlling cloud usage).
Experiments on mathematical benchmarks show the method outperforms baselines (e.g., task-tuning only, router-based offloading) in accuracy, narrows the performance gap to full cloud LLM, and maintains stable cloud usage under constraints.

**Strengths:**

- Eliminates External Router Limitations: Unlike prior two-stage pipelines (task-tuning + separate router), the framework embeds routing into the on-device LLM’s post-training. This avoids redundant computation/storage of routers, resolves the router’s inability to judge task difficulty from prompt surface patterns, and simplifies system complexity— a critical practical improvement.
- Theoretically Grounded RL Design: The GAPG algorithm addresses GRPO’s key flaws: the group-level gradient estimator  ensures unbiased optimization of the reward objective
- The method is validated across diverse on-device models (1B–3B parameters) and benchmarks. It consistently outperforms baselines, maintains accuracy gains when cloud call ratios vary, and transfers well to unseen tasks— demonstrating robustness beyond training datasets.

**Weaknesses:**

- Limited Task Scope to Mathematics: All experiments focus on mathematical reasoning tasks (arithmetic puzzles, competition math). The framework’s effectiveness for other domains (e.g., natural language understanding, code generation) is untested— especially critical since prompt design  and reward logic  may not generalize to subjective tasks.
- The method relies on historical user interaction data to train routing strategies, but it does not address cold-start for new users/on-device models (no prior task knowledge) or dynamic cloud constraints (e.g., real-time bandwidth limits, cloud cost fluctuations). These are major gaps for real-world deployment.
- Overhead of Group Sampling in GAPG: The algorithm requires sampling G responses (G=8 in experiments) per prompt for group-level gradient estimation. This increases training computation, with no analysis of how G impacts training efficiency/performance.

**Questions:**

- How would the framework adapt to non-mathematical tasks (e.g., text summarization, code debugging)? For subjective tasks where "accuracy" is hard to quantify (e.g., summary quality), would the hierarchical reward need re-design (e.g., human preference-based rewards) require domain-specific adjustments?
- Can the method be extended to cold-start scenarios (new on-device models or users with no task history)? For example, could pre-trained routing priors (e.g., from similar tasks) or auxiliary signals (e.g., item category for recommendation) initialize the on-device LLM’s routing ability before task-specific post-training?
- What is the computational-efficiency trade-off of group size G in GAPG? Does reducing G (e.g., G=4) lower training overhead without sacrificing performance?

---

> ### Author Response · Authors · 2025-11-23
> **Responses (1/2)**
>
> We thank the reviewer for the time, effort, and valuable feedback.
>
> ### **W1. Limited Task Scope: All experiments focus on mathematical reasoning tasks. The framework’s effectiveness for other domains (e.g., natural language understanding, code generation) is untested. The major concern is that prompt design and reward logic may not generalize to subjective tasks.**
>
> We agree with the reviewer that designing reward functions for subjective tasks is inherently challenging. However, this challenge is **not specific to our method**, it is a well-recognized limitation across the entire reinforcement-learning-with-verified-rewards (RLVR) family of approaches. Existing RLVR-based post-training studies (e.g., GRPO and GRPO variants) are almost exclusively evaluated on tasks with objective correctness signals, such as mathematics reasoning. Reliable reward design for open-ended tasks remains an open research problem.
>
> In this context, our work follows the same established practice. We intentionally focus on mathematical and arithmetic reasoning tasks because they provide **deterministic, verifiable reward signals**. Our contribution is to make the **first attempt** at an end-to-end optimization framework for on-device–cloud collaborative inference, with a particular focus on enabling **collaborative reasoning**. Extending collaborative device–cloud RL training to subjective tasks would require **reward-model-based evaluation** or **LLM-as-a-Judge mechanisms**, which is an important direction for future work.
>
> > The limitation noted by the reviewer is **a limitation of the RLVR regime**, not of our framework specifically. Our work inherits the same scope as other RLVR methods and can be applied to the tasks where deterministic reward signals exist. To clarify this, we revised the manuscript and added a discussion on the potential limitation of our work.
>
> ### **W2. Overhead of Group Sampling in GAPG: The algorithm requires sampling G responses (G=8 in experiments) per prompt for group-level gradient estimation. This may increase training computation, with no analysis of how G impacts training efficiency and performance.**
>
> In GAPG, the group size ($G$) controls a trade-off: too small a group may reduce the diversity of sampled responses, while overly large groups increase computation. This group sampling is inspired by GRPO. Following common practice in GRPO-style training, we adopt a moderate group size $G=8$, which provides stable optimization while keeping computation cost manageable.
>
> To address this comment, we added an experiment under varying group size based on the MATH task and Qwen2.5-1.5B-Instruct model. The results are presented below.
>
> **Qwen2.5-1.5B-Instruct**
>
> | Group size | MATH-lighteval | MATH-500 | AMC23 | AIME24 | MinervaMath | Avg. |
> | ---------- | -------------- | -------- | ----- | ------ | ----------- | ---- |
> | 4          | 56.0           | 61.3     | 35.0  | 23.3   | 22.1        | 39.5 |
> | 8          | 78.0           | 81.6     | 57.5  | 23.3   | 40.8        | 56.2 |
> | 16         | 78.2           | 81.1     | 57.5  | 23.3   | 41.2        | 56.3 |
> | 32         | 78.3           | 81.9     | 60.0  | 23.3   | 41.5        | 57.0 |
>
> **Llama-3.2-3B-Instruct**
>
> | Group size | MATH-lighteval | MATH-500 | AMC23 | AIME24 | MinervaMath | Avg. |
> | ---------- | -------------- | -------- | ----- | ------ | ----------- | ---- |
> | 4          | 53.2           | 53.9     | 27.5  | 23.3   | 33.5        | 38.3 |
> | 8          | 79.5           | 68.6     | 52.5  | 33.3   | 43.4        | 55.5 |
> | 16         | 79.4           | 69.2     | 52.5  | 33.3   | 44.1        | 55.7 |
> | 32         | 80.1           | 68.7     | 52.5  | 33.3   | 43.8        | 55.7 |
>
> As shown in the table, reducing the group size to $G=4$ leads to a clear performance drop. A potential reason is that a smaller group provides lower response diversity during sampling, increasing the risk of missing low-probability but high-quality generations. In contrast, increasing $G$ to 16 or 32 does not produce noticeable performance improvements, while introducing substantially higher computational cost. Overall, these results indicate that a moderate group size offers the best balance between performance and efficiency.
>
> > We included this ablation study in **Appendix C** of the revised manuscript.

---

> ### Author Response · Authors · 2025-11-23
> **Responses (2/2)**
>
> ### **Q1. How would the framework adapt to other tasks (e.g., text summarization, code debugging)? For subjective tasks where "accuracy" is hard to quantify (e.g., summary quality), would the hierarchical reward still benefit?**
>
> We thank the reviewer for this question. Our method is primarily applicable to tasks where the **deterministic, verifiable reward signals**  are available, similar to other reinforcement-learning-with-verified-rewards (RLVR) approaches. A promising path forward to subjective tasks is to incorporate **reward-model-based evaluation** or **LLM-as-a-Judge mechanisms**, which can provide soft quality judgments. We can set a quality threshold and assign accuracy rewards only when the generated response exceeds the threshold, while encouraging calls for help when the on-device model is uncertain about its generation, which form the hierarchical reward. Integrating such reward-model–based signals into our collaborative training framework is a natural and important direction for future work.
>
> ### **Q2. What is the computational-efficiency trade-off of group size G in GAPG? Does reducing G lower training overhead without sacrificing performance?**
>
> The group size $G$ indeed introduces a computation-performance trade-off: too small a group may reduce the diversity of sampled responses, while overly large groups increase computation. To address this comment, we evaluated the performance of our algorithm under several different group sizes in **Appendix C** of the revised manuscript . We also attached the results in our response to **W2**.
>
> Again, we appreciate the reviewer’s time and thoughtful comments.

---

### Official Review · Reviewer_6Grk · 2025-10-30

**Soundness:** 2
**Presentation:** 3
**Contribution:** 2
**Rating:** 4
**Confidence:** 3

**Summary:**

This paper proposes a unified framework for device-cloud LLM collaboration, where an on-device model is post-trained to jointly solve problems and decide when to delegate difficult queries to a cloud LLM under a budget . The authors identify that standard RL algorithms like GRPO fail on this task, as their reward normalization flattens the crucial reward hierarchy (e.g., correct answer > call for help), causing the model to learn a suboptimal, "always-delegate" policy . To fix this, they introduce the Group-Adaptive Policy Gradient (GAPG), a novel RL algorithm featuring an un-normalized policy gradient to respect the reward hierarchy and adaptive prompt filtering to enforce the cloud budget.

**Strengths:**

The core idea of embedding the routing decision within the on-device LLM is a practical and intuitive contribution. It correctly leverages the internal state of the model to inform the routing decision, which is a clear advantage over "prompt-only" routers.

The paper also provides a clear analysis of why a state-of-the-art RL algorithm like GRPO fails in this setting (Section 3.3). The identification of the "misalignment of normalized advantage" (where high-priority sparse rewards and low-priority dense rewards are treated as equal post-normalization) is a sharp insight that provides a solid motivation for their proposed algorithm.

**Weaknesses:**

Limited Task Diversity: The experiments are exclusively focused on mathematical and arithmetic reasoning tasks. These tasks are well-suited for this problem as they have clear, binary correctness signals. However, the paper's central claim is about a general collaborative framework. It is unclear how this method would perform on more open-ended, creative, or subjective tasks (e.g., summarization, dialogue, writing) where defining a simple, non-binary reward signal r(x,y) is a significant challenge in itself.

Limited Comparison to Suitable RL Alternatives: While the analysis of GRPO's failure is sound, the paper does not justify why a new algorithm (GAPG) is necessary over other, more established RL frameworks. The problem is formulated as a classic constrained, multi-objective problem. Furthermore, the paper omits comparisons to entire classes of algorithms designed for this setup, such as Multi-Objective RL (MORL) (e.g., Hayes et al., 2022), which are built to handle competing objectives. Moreover, comparison with DPO and CPO methods is missing.

Comparison with the router-based approach: The paper's experimental comparison to the "Task-Tuning&Router" baseline is weak, as it appears to compare against a "prompt-only" router. The authors' main critique—that it is difficult to assess task difficulty from the prompt alone—is valid, but this is a known limitation. A much stronger and more practical baseline would be a "post-inference" router that takes the on-device LLM's full response and/or confidence as input. This type of "quality-gap" predictor, which is also mentioned in the work they cite (Ding et al., 2024) would serve as a much stronger and more relevant baseline for comparison.


[CF Hayes, R Rădulescu, et al. A Practical Guide to Multi-Objective Reinforcement Learning and Planning]

[D Ding et al. Hybrid LLM: Cost-Efficient and Quality-Aware Query Routing]

**Questions:**

1. In Figure 5, the "call-for-cloud ratio" is treated as an independent variable. Could you clarify how this graph was generated for the "Collaboration-Aware Tuning" (GRPO) baseline? Given the claim in Section 3.3 that this method collapses to an "always-call-for-cloud" policy, how was it forced to operate under lower, fixed budgets (e.g., 20%) at test time?

2. The proposed GAPG algorithm still requires calls to the cloud LLM during training (Algorithm 1, Step 6) to compute the coordination reward. This introduces a significant training cost. The x-axis in Figures 3 & 4 (training iterations) does not capture this. Could you provide an analysis of the total training cost (e.g., number of cloud calls) required by GAPG versus the baselines (including the two-stage cost of training the router) to achieve the final reported accuracy?

3. The 'Task-Tuning&Router' baseline is presented as a single final performance point (⋆) in Figures 3 and 4. To help verify that this baseline was trained sufficiently and to understand its learning dynamics, could you provide a plot showing this baseline's accuracy as a function of its router-training iterations?

---

> ### Author Response · Authors · 2025-11-24
> **Responses (1/3)**
>
> We thank the reviewer for the time, effort, and valuable feedback.
>
> ### **W1. Limited Task Diversity: The experiments are exclusively focused on mathematical and arithmetic reasoning tasks. These tasks are well-suited for this problem as they have clear, binary correctness signals. However, the paper's central claim is about a general collaborative framework. It is unclear how this method would perform on more open-ended, creative, or subjective tasks (e.g., summarization, dialogue, writing) where defining a simple, non-binary reward signal r(x,y) is a significant challenge in itself.**
>
> We agree with the reviewer that designing reward functions for open-ended or subjective tasks is inherently challenging. However, this challenge is **not specific to our method**, it is a well-recognized limitation across the entire reinforcement-learning-with-verified-rewards (RLVR) family of approaches. Existing RLVR-based post-training studies (e.g., GRPO and GRPO variants) are almost exclusively evaluated on tasks with objective correctness signals, such as mathematics reasoning. Reliable reward design for open-ended tasks remains an open research problem.
>
> In this context, our work follows the same established practice. We intentionally focus on mathematical and arithmetic reasoning tasks because they provide **deterministic, verifiable reward signals**. Our contribution is to make the **first attempt** at an end-to-end optimization framework for on-device–cloud collaborative inference, with a particular focus on enabling **collaborative reasoning**. Extending collaborative device–cloud RL training to more complex or subjective tasks would require **reward-model-based evaluation** or **LLM-as-a-Judge mechanisms**, which is an important direction for future work.
>
> > The limitation noted by the reviewer is **a limitation of the RLVR regime**, not of our framework specifically. Our method inherits the same scope as other RLVR methods and can be applied to the tasks where deterministic reward signals exist. To clarify this, we revised the manuscript and added a discussion on the potential limitation of our work.
>
> ### **W2. Limited Comparison to Suitable RL Alternatives. The paper omits comparisons to entire classes of algorithms designed for this setup, such as Multi-Objective RL, DPO, and CPO methods. Why a new algorithm (GAPG) is necessary over other, more established RL?**
>
> **Comparison to Suitable RL Alternatives**
>
> We apologize for the misunderstanding regarding the role of multi-objective optimization in our formulation. Our problem involves two aspects, coordination reward and accuracy reward, these are **not** treated as independent objectives as in conventional Multi-Objective RL (MORL). In MORL, objectives form a multi-dimensional objective vector that requires an explicit aggregation or scalarization step before optimization, as discussed in the paper cited by the reviewer. In contrast, in our work the hierarchical feedback is **already aggregated into a single scalar reward**. Therefore, MORL formulations and the associated aggregation strategies are not applicable to our setting.
>
> **DPO and CPO** are also not applicable. Both are designed for **offline RL with preference pairs** and rely on datasets containing preference comparisons. Such data **does not exist in our setting**, especially lacking the data guilding the model to ask help from the cloud. Moreover, these algrorithm do not consider the specific requirements in our device-cloud collaboration setting, such as hierarchical reward and controlling the call-for-help ratio.
>
> **Why a new algorithm (GAPG) is necessary over other, more established RL?**
>
> In our work, we **formulate, for the first time, a device-cloud collaborative optimization problem** in which the call-for-help ratio must be explicitly controlled, as reflected in the constraint of Problem (2). In the LLM fine-tuning literature, we didn't see prior work that models or optimizes this collaboration behavior in a principled manner. In addition, the **conventional RL** works do not account for the specific requirements of LLM fine-tuning. This gap motivates the development of our **GAPG algorithm**, which is specifically designed to handle the device–cloud coordination constraint while maintaining stable RL-based optimization.

---

> ### Author Response · Authors · 2025-11-24
> **Responses (2/3)**
>
> ### **W3. The paper's experimental comparison to the "Task-Tuning&Router" baseline is weak. A much stronger and more practical baseline would be a "post-inference" router that takes the on-device LLM's full response and/or confidence as input. This type of "quality-gap" predictor, which is also mentioned in the work they cite (Ding et al., 2024) would serve as a much stronger and more relevant baseline for comparison.**
>
> We would like to clarify that the *Task-Tuning&Router* baseline we use is **exactly the method proposed in Ding et al. (2024)**. In that work, as in our implementation, the router takes the **query** as input, not the full on-device LLM response. The generated response from the small model is used only to derive the *label* indicating the quality that the small model can provide, which is **not** fed into the router during inference. Therefore, **our baseline faithfully follows the design of Ding et al. (2024)**. Ding et al. used a BART score to measure the response quality and thus proposed *quality-gap predictor*. In contrast, our tasks have clear **correctness** signals, so no additional quality-gap modeling is needed.
>
> To address this concern, we also report the performance of a post-inference router. As shown in the table, it yields a only marginal improvement but remains clearly inferior to our proposed methods. The potential reason is that even if the response is inputed, the lightweight binary classifier typically lacks the ability to check whether there are reasoning errors in the response.
>
> **Qwen2.5-1.5B-Instruct**
>
> | Method                | MATH-lighteval | MATH-500 | AMC23 | AIME24 | MinervaMath | Avg. |
> | --------------------- | -------------- | -------- | ----- | ------ | ----------- | ---- |
> | Pre-inference router  | 70.3           | 71.6     | 55.0  | 23.3   | 34.9        | 51.0 |
> | Post-inference router | 70.9           | 72.2     | 55.0  | 23.3   | 36.8        | 51.6 |
> | Ours                  | 78.0           | 81.6     | 57.5  | 23.3   | 40.8        | 56.2 |
>
> **Llama-3.2-3B-Instruct**
>
> | Method               | MATH-lighteval | MATH-500 | AMC23 | AIME24 | MinervaMath | Avg. |
> | -------------------- | -------------- | -------- | ----- | ------ | ----------- | ---- |
> | Pre-inference router | 68.7           | 59.6     | 42.5  | 30.0   | 36.4        | 47.4 |
> | Pre-inference router | 69.4           | 61.0     | 45.0  | 30.0   | 37.1        | 48.5 |
> | Ours                 | 79.5           | 68.6     | 52.5  | 33.3   | 43.4        | 55.5 |
>
> > We have already replaced the performance of the router based scheme with the post inference router and updated the description for this baseline.

---

> ### Author Response · Authors · 2025-11-24
> **Responses (3/3)**
>
> ### **Q1. Could you clarify how Figure 5 was generated for the "Collaboration-Aware Tuning" (GRPO) baseline?**
>
> We thank the reviewer for this detailed question. In the Countdown task, GRPO collapses into an “always-call-for-cloud’’ policy. In the MATH task (Figure 5), GRPO converges to a policy whose call-for-cloud ratio is approximately **59.7%**. When we evaluate GRPO at lower call-for-cloud ratios (e.g., $20 \%$ or $40 \%$), we truncate the extra calls beyond the target ratio. This is how the accuracies at those intermediate ratios are computed. For the two extreme ratios ($0 \%$ and $100 \%$), we report the accuracy of the local model alone and the accuracy of the cloud LLM (withour coordination), respectively.
>
> > We have added the above description of the Collaboration-Aware Tuning baseline to the explanation of Figure 5 in the revised manuscript.
>
> ### **Q2. The x-axis in Figures 3 & 4 (training iterations) does not capture the training cost. Could you provide an analysis of the total training cost (e.g., number of cloud calls) required by GAPG versus the baselines?**
>
> We would like to clarify that, as stated in Step 6 of Algorithm 1, the cloud LLM is called **at most once per prompt** during training. In our implementation, we **store the cloud LLM’s response the first time a prompt invokes a cloud call**, and this stored result is **reused** in all subsequent training iterations. Therefore, each prompt incurs at most a **single** cloud call. Overall, the cloud-call cost is bounded by the size of the training set.
>
> - **Task-Tuning & Router** first asks the on-device LLM to solve each prompt. A cloud call is made only when the on-device answer is incorrect, and this information is stored for training the router.
>
> -  **Ours (GAPG)** and **Collaboration-Aware Tuning (GRPO)** both may invoke the cloud LLM once per prompt and cache the output for reuse.
>
> Below we report the **actual number of cloud calls** incurred in training for Figures 3 and 4. The baselines Task-Tuning Only and Task-Tuning \& Naive Offloading do not use the cloud LLM and thus have zero cost, which are thus omitted from this table.
>
> | Methods  | Ours | Task-Tuning & Router | Collaboration-Aware Tuning |
> | -------- | ---- | -------------------- | -------------------------- |
> | Figure 3 | 2519 | 2540                 | 6000                       |
> | Figure 4 | 2883 | 3075                 | 6746                       |
>
> As shown in the table, among the baselines that invoke the cloud LLM during training, our method **does not** incur a higher cloud-call cost.
>
> > We have added the above discussion and the empirical comparison to **Appendix H** of the revised manuscript
>
> ### **Q3. Provide a plot showing this baseline's accuracy versus router-training iterations**
>
> We thank the reviewer for this comment. To address this, we added a section **Appendix G** in the revised manuscript to show the learning dynamics of the router and to present the accuracy of router based baseline versus training iterations of the router.
>
> Again, we appreciate the reviewer’s time and thoughtful comments.

---

> ### Comment · Reviewer_6Grk · 2025-11-28
>
> Thank you for your very detailed and helpful response. I still have 2 concerns:
>
> 1. Reduced Contribution Scope (Admission of Similarity to Dr. GRPO):
> In the rebuttal, you acknowledged the strong similarity between your proposed GAPG algorithm and the existing "Dr. GRPO" method, clarifying that your primary contribution is not the algorithm itself, but rather the application of this method to the device-cloud setting. This concession significantly weakens the paper's claimed novelty.
>
>
> 2. Misrepresentation of the Router Baseline's Scope:
> I disagree with the argument that experiments on broader domains (like summarization) are impossible because "data does not exist." As authors mentioned, this limitation applies to RLVR methods (which require online, programmatic verification). The Router baseline is a standard classifier that can be easily trained on existing offline preference datasets.
> By restricting the comparison to only verifiable domains (Math), you are masking a major advantage of the Router approach: its applicability to real-world, non-verifiable tasks. This questions the practical use of the method as it does not seem to be a general replacement for routing, but a specialized solution for narrow, verifiable domains.
>
>
> Given these concerns, I'm maintaining my score.

---

> ### Author Response · Authors · 2025-11-29
>
> Thank you for the reply. To clarify:
>
> (1) Our distinctions with Dr.GRPO lie in **adaptive filtering**, **hierarchical rewards**, and an **online algorithm**, which are included in Appendix B.
>
> (2) Our statement that ``data does not exist'' refers to **DPO/CPO-style preference methods**, not to **router-based classifiers**.
>
> (3) We do not claim GAPG is a general replacement for routers. Instead, GAPG addresses a setting that existing router methods performs poorly, **reasoning problem**, serving as a strong **complement** to the router-based method.

---

### Author Response · Authors · 2025-12-03
**Summary**

Dear Area Chair,

We greatly appreciate your time and effort in handling our submission. At this stage, we would like to offer a brief summary of the reviewers' main concerns and how we addressed them in our rebuttal.

The main comments provided by the reviewers were:

1. What's our contribution beyond GRPO?
2. How sensitive are the hierarchical reward weights and the group size?
3. Can you extend experiments to subjective tasks?

In our response, we clarified the following:

1. Our contributions in this work are three-fold:

**Collaboration paradigm.** We propose the first principled approach to integrate routing behavior *directly* into the on-device LLM through RL-based post-training, enabling the model to learn when to rely on its own reasoning and when to offload to the cloud LLM.

**Problem formulation.** We introduce a hierarchical reward design that jointly captures task correctness and device-cloud coordination. This formulation is unique to the collaborative inference setting and is essential for this methodology.

**Algorithmic design.** We develop a policy-gradient method tailored to this formulation, featuring **group-response sampling** (inspired by GRPO but not equivalent to it and its variants, Dr. GRPO) and **adaptive prompt filtering**, both of which are specific to our collaboration inference problem.

2. We added ablation study on the hierarchical reward weights and the group size in Appendix C.
3. We intentionally focused on **reasoning tasks** because they provide **verifiable reward signals**. The limitation noted by the reviewer is **a limitation of the RLVR regime**, not of our framework specifically. Our method inherits the same scope as other RLVR methods and can be applied to the tasks where deterministic reward signals exist. Extending it to subjective tasks would require **reward-model-based evaluation** or **LLM-as-a-Judge mechanisms**, which is an important direction for future work.

Although the reviewers did not follow up from the reviewers (Reviewer faeX updated his review at **Nov 17** and **increased his score to 6**, we believe their concerns were adequately addressed in our rebuttal.

Thank you again for your coordination.

Best regards,

**Authors**

---

### Meta-Review · Area_Chair_RK2p · 2026-01-07

**Summary:**

The paper proposes a collaborative device-cloud inference framework where the on-device Large Language Model (LLM) autonomously decides to offload queries via a hierarchical reward system and a Group-Adaptive Policy Gradient (GAPG) algorithm.

While the goal of integrating routing capabilities directly into the on-device model is practical, however, the limited novelty of the optimization algorithm (due to its similarity to Dr. GRPO), the restriction to math-only tasks, and the valid concerns regarding experiment scope and inference efficiency are not fully resolved in the rebuttal, indicating the paper is not yet ready for publication at ICLR. Thus, the decision is reject.

**Reviewer Concerns:**

The limited novelty of the optimization algorithm (due to its similarity to Dr. GRPO), the restriction to math-only tasks, and the valid concerns regarding experiment scope and inference efficiency are not fully resolved in the rebuttal.

**Reviewer Scores:**

The reviewer scores are 4442 which I do not think would change.

---

### Decision · Program_Chairs · 2026-01-26

Reject